# CONFIDENCE-DRIVEN SAMPLING FOR BACKDOOR ATTACKS

## ABSTRACT

Backdoor attacks aim to surreptitiously insert malicious triggers into DNN models, granting unauthorized control during testing scenarios. Existing methods lack robustness against defense strategies and predominantly focus on enhancing trigger stealthiness while randomly selecting poisoned samples. Our research highlights the overlooked drawbacks of random sampling, which make that attack detectable and defensible. The core idea of this paper is to strategically poison samples near the model's decision boundary and increase defense difficulty. We introduce a straightforward yet highly effective sampling methodology that leverages confidence scores. Specifically, it selects samples with lower confidence scores, significantly increasing the challenge for defenders in identifying and countering these attacks. Importantly, our method operates independently of existing trigger designs, providing versatility and compatibility with various backdoor attack techniques. We substantiate the effectiveness of our approach through a comprehensive set of empirical experiments, demonstrating its potential to significantly enhance resilience against backdoor attacks in DNNs.

## 1 INTRODUCTION

During DNN training on large datasets or third-party collaborations, there exist concerns about potential malicious triggers injected into the model. These intrusions can lead to unauthorized manipulation of the model's outputs during testing, causing what is commonly referred to as a "backdoor" attack (Li et al., 2022; Doan et al., 2021a). To elaborate, attackers can inject triggers into a small portion of training data in a specific manner. Attackers may then provide either the poisoned training data or backdoored models trained on it to third-party users, depending on their capabilities (Li et al., 2022). In the inference stage, the injected backdoors are activated via triggers, causing triggered inputs to be misclassified as a target label. To date, numerous backdoor attack methods, such as BadNets (Gu et al., 2017), WaNet (Nguyen & Tran, 2021), label-consistent (Turner et al., 2019), have demonstrated strong attack performance. These methods consistently achieve high attack success rates while maintaining a high accuracy on clean data within mainstream DNNs.

An important research direction in backdoor attacks is to enhance of the stealthiness of poisoned samples while concurrently ensuring their effectiveness. Most efforts in this research line have been made to trigger design (e.g., hidden triggers Saha et al., 2020, clean-label (Turner et al., 2019)). However, in the vast majority of existing attack methods, samples are randomly chosen from the clean training set for poisoning. However, our preliminary study (in Section 4.1) observes that the "random sampling" strategy is highly related to the possibility of the poisoning samples to be detected by the defenders. Moreover, it is totally practical and feasible for the attackers to choose the poisoning samples from the training set with preference. In fact, it is a common setting to assume that the attacker has the knowledge of the victim model's training dataset for sample selection. For example, the victim models can be trained on downloaded online datasets, which are provided by the attacker (Li et al., 2022). Similarly, the attacker can also act as model providers to directly provide the backdoored models (Nguyen & Tran, 2021; Doan et al., 2021b). Therefore, there is plenty of room for discussing the sampling strategies in this scenario, which raises the question: *Is there a better sampling strategy to improve the stealthiness of backdoors?*

To answer this question, in Section 4.1, we first take a closer look at the random sampling strategy, by investigating the latent space of the backdoored model. From the visualizations in Figure 1, we draw two interesting findings: First, most of the randomly chosen samples are close to the center of their true classes in the latent space; second, the closer a sample is from its true class on the clean

model, the further it gets from the target class on the backdoored model. These two observations reveal important clues about the "stealthiness" of the random sampling strategy, which suggest the randomly sampled data points may cause them to be easier to be detected as outliers. To have a deeper understanding, we further build a theoretical analysis of SVM in the latent space (Section 4.3) to demonstrate the relation between the random sampling strategy and attack stealthiness. Moreover, our observations suggest an alternative to the random sampling—it is better to select samples that are closer to the decision boundary. Our preliminary studies show that these **boundary samples** can be manipulated to be closer to the clean samples from the target class, and can greatly enhance their stealthiness under potenital outlier detections (see Figure 1c and 1d).

Inspired by these discoveries, we propose a novel method called **confidence-driven boundary sampling** (CBS). To be more specific, we identify boundary samples with low confidence scores based on a surrogate model trained on the clean training set. Intuitively, samples with lower confidence scores are closer to the boundary between their own class and the target class in the latent space (Karimi et al., 2019) and can avoid vulnerabilities brought by random samplings. Therefore, this strategy makes it more challenging to detect attacks. Moreover, our sampling strategy is independent from existing attacking approaches which makes it exceptionally versatile. It easily integrates with a variety of backdoor attacks, offering researchers and practitioners a powerful tool to enhance the stealthiness of backdoor attacks without requiring extensive modifications to their existing methods or frameworks. Extensive experiments combining proposed confidence-based boundary sampling with various backdoor attacks illustrate the advantage of the proposed method compared with random sampling.

## 2 RELATED WORKS

### 2.1 BACKDOOR ATTACKS AND DEFENSES

As mentioned in the introduction, backdoor attacks are shown to be a serious threat to deep neural networks. BadNet (Gu et al., 2017) is the first exploration that attaches a small patch to samples and therefore introduces backdoors into a DNN model. After that many efforts are put into developing advanced attacks to either boost the performance or improve the resistance against potential defenses. Various trigger designs are proposed, including image blending (Chen et al., 2017), image warpping (Nguyen & Tran, 2021), invisible triggers (Li et al., 2020; Saha et al., 2020; Doan et al., 2021b), clean-label attacks (Turner et al., 2019; Saha et al., 2020), sample-specific triggers (Li et al., 2021b; Souri et al., 2022), etc. These attacking methods have demonstrated strong attack performance (Wu et al., 2022). In the meanwhile, the study of effective defenses against these attacks also remains active. One popular type of defense depends on detecting outliers in the latent space (Tran et al., 2018; Chen et al., 2018; Hayase et al., 2021; Gao et al., 2019; Chen et al., 2018). Other defenses incorporate neuron pruning (Wang et al., 2019), detecting abnormal labels (Li et al., 2021a), model pruning (Liu et al., 2018), fine-tuing (Sha et al., 2022), etc.

### 2.2 SAMPLINGS IN BACKDOOR ATTACKS

While the development of triggers in backdoor attacks attracts much attention, the impact of poisoned sample selections is rarely explored. As far as we know, Xia et al. (2022) is the only work focusing on the sampling method in backdoor attacks. They proposed a filtering-and-updating strategy (FUS) to select samples with higher contributions to the injection of backdoors by computing the forgetting event (Toneva et al., 2018) of each sample. For each iteration, poison samples with low forgetting events will be removed and new samples will be randomly sampled to fill up the poisoned training set. Though this method shows improvement in performance, it ignores the backdoor's ability to resist defenses, known as the 'stealthiness' of backdoors. To the best of our knowledge, we are the first work study the stealthiness problem from the perspective of sampling.

## 3 DEFINITION AND NOTATION

In this section, we will introduce preliminaries about backdoor attacks, including the threat model discussed in this paper and a general pipeline that is applicable to many backdoor attacks.

### 3.1 THREAT MODEL

We follow the commonly used threat model for the backdoor attacks (Gu et al., 2017; Doan et al., 2021b). We assume that the attacker has access to the clean training set and can modify any subset from the training data. Then the victim trains his own models on this data and the attacker has no knowledge of this training procedure. In a real-world situation, attackers can upload their datasets to

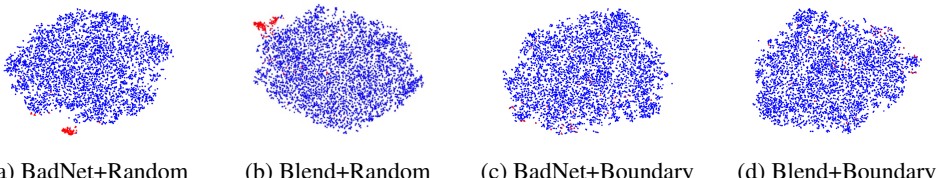

(a) BadNet+Random     (b) Blend+Random     (c) BadNet+Boundary     (d) Blend+Boundary

Figure 1: Latent space visualization of BadNet and Blend via **Random** and **Boundary** sampling.

the internet. They can sneakily insert backdoors into their data and then share it with victims, who unknowingly use it to train their own models (Gu et al., 2017; Chen et al., 2017). Note that many existing backdoor attacks (Nguyen & Tran, 2021; Turner et al., 2019; Saha et al., 2020) already adopt this assumption and our proposed method does not demand additional capabilities from attackers beyond what is already assumed in the context of existing attack scenarios.

### 3.2 A GENERAL PIPELINE FOR BACKDOOR ATTACKS

In the following, we introduce a general pipeline, which is applicable to a wide range of backdoor attacks. The pipeline consists of two components.

**(1) Poison sampling**. Let $D_{tr} = \{(x_i, y_i)\}_{i=1}^n$ denote the set of $n$ clean training samples, where $x_i \in \mathcal{X}$ is each individual input sample with $y_i \in \mathcal{Y}$ as the true class. The attacker selects a subset of data $U \subset D_{tr}$, with $p = |U|/|D_{tr}|$ as the poison rate, where the poison rate $p$ is usually small.

**(2) Trigger injection**. Attackers design some strategies $T$ to inject the trigger $t$ into samples selected in the first step. In specific, given a subset of data $U$, attackers generate a poisoned set $T(U)$ as:

$$T(U) = \{(x', y')|x' = G_t(x), y' = S(x, y), \forall (x, y) \in U\} \tag{1}$$

where $G_t(x)$ is the attacker-specified poisoned image generator with trigger pattern $t$ and $S$ indicates the attacker-specified target label generator. After training the backdoored model $f(\cdot; \theta^b)$ on the poisoned set, the injected backdoor will be activated by trigger $t$. For any given clean test set $D_{te}$, the accuracy of $f(\cdot; \theta^b)$ evaluated on trigger-embedded dataset $T(D_{te})$ is referred to as success rate, and attackers expect to see high success rate on any clean samples with triggers embedded.

## 4 METHOD

In this section, we will first analyze the commonly used random samplings, and then introduce our propose method as well as some theoretical understandings.

### 4.1 REVISIT RANDOM SAMPLING

**Visualization of Stealthiness.** Random sampling selects samples to be poisoned from the clean training set with the same probability and is commonly used in existing attacking methods. However, we suspect that such unconstrained random sampling is easy to be detected as outliers of the target class in the latent space. To take a look at the sample distribution in latent space, we first conduct TSNE (Van der Maaten & Hinton, 2008) visualizations of clean samples from the target class, and the poisoned samples which could be originally from other class but labeled as the target class. We consider these poisoned samples are obtained by two representative attack algorithms, BadNet (Gu et al., 2017) and Blend (Chen et al., 2017) both of which apply random sampling, on CIFAR10 (Krizhevsky et al., 2009), in Figure 1a and 1b. In detail, the visualizations show the latent representations of samples from the target class, and the colors red and blue indicate poisoned and clean samples respectively. It is obvious that there exists a clear gap between poisoned and clean samples. For both attacks, most of the poisoned samples form a distinct cluster outside the clean samples. This will result in separations in latent space which can be easily detected by possible defenses. For example, Spectral Siginiture (Tran et al., 2018), SPECTRE (Hayase et al., 2021), SCAn (Tang et al., 2021) are representative defenses relying on detecting outliers in the latent space and show great power defending various backdoor attcks (Wu et al., 2022).

**Relation between Stealthiness & Random Sampling.** In our study, we also observe the potential relation between random sampling and the stealthiness of backdoors. To elaborate, we further calculate the distance from each selected sample (without trigger) to the center[1] of their true classes

---

[1]The average of sample representations within this class.

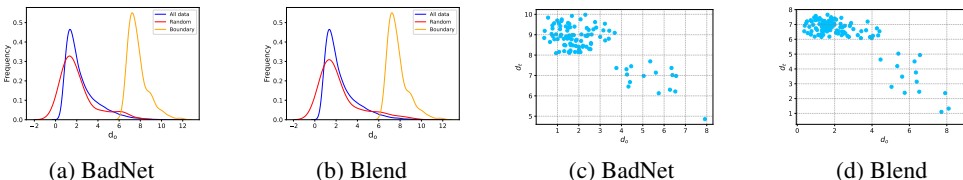

| (a) BadNet | (b) Blend | (c) BadNet | (d) Blend |

Figure 2: The left two figures depict the distribution of $d_o$ when samples are Randomly selected by BadNet and Blend. The right two figures shows the relationship between $d_o$ and $d_t$ for BadNet and Blend.

computed on the clean model, which is denoted as $d_o$. As seen in Figure 2a and 2b, random sampling tends to favor samples that are close to their true classes. However, we find $d_o$ may have an obvious correlation with the distance between the sample and the target class which we visualize in the previous Figure 1. Formally, we define the distance between each selected sample (with trigger) and the center of the target class computed on the backdoored model, as $d_t$. From Figure 2c and 2d, we observe a negative correlation between $d_t$ and $d_o$, indicating that samples closer to the center of their true classes in the clean model tend to be farther from the target class after poisoning and thus easier to detect. These findings imply that random sampling often results in the selection of samples with weaker stealthiness. Our observations also suggest that samples closer to the boundary may lead to better stealthiness, and motivate our proposed method.

## 4.2 CONFIDENCE-DRIVEN BOUNDARY SAMPLING (CBS)

One key challenge for boundary sampling is how to determine which samples are around the boundaries. Though we can directly compute the distance from each sample to the center of the target class in the latent space and choose those with smaller distances, this approach can be time-consuming, as one needs to compute the center of the target class first and then compute the distance for each sample. This problem can be more severe when the dataset's size and dimensionality grow. Consequently, a more efficient and effective method is in pursuit.

To solve this issue, we consider the *confidence score*. To be more specific, we inherit the notations from Section 3.2 and further assume there exist $K$ classes, i.e. $\mathcal{Y} = \{1, ..., K\}$, for simplicity. Let $f(\cdot; \theta)$ denote a classifier with model parameter $\theta$, and the output of its last layer is a vector $z \in \mathbb{R}^K$. *Confidence score* is calculated by applying the softmax function on the vector $z$, i.e. $s_c(f(x; \theta)) = \sigma(z) \in [0, 1]^K$, where $\sigma(\cdot)$ is the softmax function. This confidence score is considered the most accessible uncertainty estimate for deep neural network (Pearce et al., 2021), and is shown to be closely related to the decision boundary (Li et al., 2018; Fawzi et al., 2018). Since our primary goal is to identify samples that are closer to the decision boundary, we anticipate finding samples with similar confidence for both the true class[2] and the target class. Thus, we can define boundary samples as:

**Definition 4.1** (**Confidence-based boundary samples**). *Given a data pair $(x, y)$, model $f(\cdot; \theta)$, a confidence threshold $\epsilon$ and a target class $y'$, if*

$$|s_c(f(x; \theta))_y - s_c(f(x; \theta))_{y'}| \leq \epsilon \qquad (2)$$

*Then $(x, y)$ is noted as $\epsilon$-boundary sample with target $y'$.*

To explain Definition 4.1, since $s_c(f(x; \theta))_y$ represents the probability of classifying $x$ as class $y$, then when there exists another class $y'$, for which $s_c(f(x; \theta))_{y'} \approx s_c(f(x; \theta))_y$, it signifies that the model is uncertain about whether to classify $x$ as class $y$ or class $y'$. This uncertainty suggests that the sample is positioned near the boundary that separates class $y$ from class $y'$ (Karimi et al., 2019).

The proposed **Confidence-driven boundary sampling** (CBS) method is based on Definition 4.1. In general, CBS selects boundary samples in Definition 4.1 for a given threshold $\epsilon$. Since we assume the attacker has no knowledge of the victim's model, we apply a surrogate model like what black-box adversarial attacks often do (Chakraborty et al., 2018). In detail, a pre-trained surrogate model $f(\cdot; \theta)$ is leveraged to estimate confidence scores for each sample, and $\epsilon$-boundary samples with pre-specified target $y^t$ are selected for poisoning. The detailed algorithm is shown in Algorithm 1 in Appendix A.3. It is worth noting that the threshold $\epsilon$ is closely related to poison rate $p$ in Section 3.2, and we can determine $\epsilon$ based on $|U(y^t, \epsilon)| = p \times |\mathcal{D}_{tr}|$. Since we claim that our sampling method can be easily adapted to various backdoor attacks, we provide an example that adapts our sampling methods to Blend (Chen et al., 2017), where we first select samples to be poisoned via Algorithm 1

---

[2]For a correctly classified sample, the true class possesses the largest score.

and then blend these samples with the trigger pattern $t$ to generate the poisoned training set. Detailed algorithms can be found in Algorithm 2 in Appendix A.3.

### 4.3 THEORETICAL UNDERSTANDINGS

To better understand CBS, we conduct theoretical analysis on a simple SVM model. As shown in Figure 3, we consider a binary classification task where two classes are uniformly distributed in two balls centered at $\mu_1$(orange circle) and $\mu_2$(blue circle) with radius $r$ respectively in latent space[3]:

$$C_1 \sim p_1(x) = \frac{1}{\pi r^2} 1[\|x - \mu_1\|_2 \leq r], \text{ and } C_2 \sim p_2(x) = \frac{1}{\pi r^2} 1[\|x - \mu_2\|_2 \leq r], \quad (3)$$

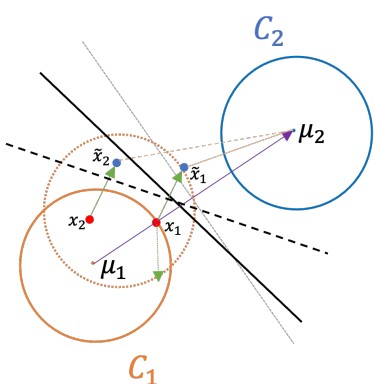

where let $\mu_2 = 0$ for simplicity. Assume that each class contains $n$ samples. We consider a simple attack that selects one single sample $x$ from class $C_1$, add a trigger to it to generate a poisoned $\tilde{x}$, and assign a label as class $C_2$ for it. Let $\tilde{C}_1, \tilde{C}_2$ denote the poisoned data, and we can obtain a new backdoored decision boundary of SVM on the poisoned data. To study the backdoor effect of the trigger, we assume $\tilde{x} = x + \epsilon \frac{t}{\|t\|}$ where $\frac{t}{\|t\|}, \epsilon$ denote the direction and strength of the trigger, respectively. To explain this design, we assume that the trigger introduces a 'feature' to the original samples (Khaddaj et al., 2023), and this 'feature' is closely related to the target class while nearly orthogonal to the prediction features[4]. In addition, we assume $t$ is fixed for simplicity, which means this trigger is universal and we argue that this is valid because existing attacks such as BadNet (Gu et al., 2017) and Blend (Chen et al., 2017) inject the same trigger to every sample. To ensure the backdoor effect, we further assume $(\mu_2 - \mu_1)^T t \geq 0$, otherwise the poisoned sample will be even further from the target class (shown as the direction of the green dashed arrow) and lead to subtle backdoor effects. We are interested in two questions: Are boundary samples harder to detect? How do samples affect the backdoor performance?

Figure 3: Backdoor on SVM

To investigate the first question, we adopt the Mahalanobis distance (Mahalanobis, 2018) between the poisoned sample $\tilde{x}$ and the target class $\tilde{C}_2$ as an indicator of outliers. A smaller distance means $\tilde{x}$ is less likely to be an outlier, indicating better stealthiness. For the second question, we estimate the success rate by estimating the volume (or area in 2D data) of the shifted class $C_1$ to the right of the backdoored decision boundary. This is because when triggers are added to every sample, the whole class will shift in the direction of $t$, shown as the orange dashed circle in Figure 3. The following theorem provides an estimation of Mahalanobis distance and success rate.

**Theorem 4.1.** *Assume $\tilde{x} = x + \epsilon t/\|t\|_2 := x + a$ for some trigger $t$ and strength $\epsilon$, and assume $(\mu_2 - \mu_1)^T t \geq 0$. Mahalanobis distance between the poisoned sample $\tilde{x}$ and the $\tilde{C}_2$ is*

$$d_M^2(\tilde{x}, \tilde{C}_2) = \frac{4n}{(n+1)r^2} \frac{(n+1)r^2}{(n+1)r^2/\|\tilde{x}\|_2^2 + 4}. \quad (4)$$

*In addition, when training the poisoned data set using the vanilla clean SVM, taking small attack strength $\epsilon$, the success rate is an increasing function of*

$$sr(\tilde{x}) = \epsilon \cos(t, \tilde{x} - \mu_1) - \|\tilde{x} - \mu_1\|/2 - r/2$$

Detailed proof can be found in Appendix A.1. Based on the theorem, a smaller $|\tilde{x}|_2$ results in a smaller $d_M^2$, making it less likely to be detected as an outlier. Additionally, a closer proximity between $\tilde{x}$ and $\mu_1$ corresponds to a higher success rate. To be more specific, we take two samples, $x_1$ close to the clean boundary (the grey dashed line), $x_2$ far from the boundary, as examples. It is obvious that $\tilde{x}_1$ is closer to center $\mu_2$, thus $\|\tilde{x}_1\| \leq \|\tilde{x}_2\|$, and then $d_M^2(\tilde{x}_1, \tilde{C}_2) \leq d_M^2(\tilde{x}_2, \tilde{C}_2)$ indicating that $\tilde{x}_1$ is harder to detect. On the other hand, as $\tilde{x}_2$ is closer to $\mu_1$, $sr(\tilde{x}_2) \geq sr(\tilde{x}_1)$, meaning boundary samples will have worse backdoor effect without defenses. These observations imply the trade-off between stealthiness and backdoor performance without defenses, and our experiments in Section 5 further illustrate that incorporating boundary samples can significantly improve the stealthiness with a slight sacrifice of success rate without defenses.

---
[3]This analysis is suitable for any neural networks whose last layer is a fully connected layer.
[4]Prediction feature here is referred to features used for prediction when no triggers invloved.

## 5 EXPERIMENT

In this section, we conduct experiments to validate the effectiveness of CBS, and show its ability to boost the stealthiness of various existing attacks. We evaluate CBS and baseline samplings under no-defense and various representative defenses in Section 5.2, 5.3, and 5.4. In Section 5.5, we will provide more empirical evidence to illustrate that CBS is harder to detect and mitigate.

### 5.1 EXPERIMENTAL SETTINGS

To comprehensively evaluate CBS and show its ability to be applied to various kinds of attacks, we consider 3 types [5] of attacking methods that cover most of existing backdoor attacks.

In detail, **Type I** backdoor attacks allow attackers to inject triggers into a proportion of training data and release the poisoned data to the public. Victims train models on them from scratch. The attack's goal is to misclassify samples with triggers as the pre-specified target class (also known as the all-to-one scenario). **Type II** backdoor attacks are similar to Type I and the difference is that victims finetune pre-trained models on poisoned data and the adversary's goal is to misclassify samples from one specific class with triggers as the pre-specified target class (also known as the one-to-one scenario). **Type III** backdoor attacks are slightly different, and allow attackers to optimize the triggers and model parameters at the same time under the all-to-one scenario.

**Baselines for sampling**. We compare CBS with two baselines—Random and FUS (Xia et al., 2022). The former selects samples to be poisoned with a uniform distribution, and the latter selects samples that contribute more to the backdoor injection via computing the forgetting events (Toneva et al., 2018) for each sample. In our evaluation, we focus on image classification tasks on datasets Cifar10 and Cifar100 (Krizhevsky et al., 2009), and model architectures ResNet18 (He et al., 2016), VGG16 (Simonyan & Zisserman, 2014). We use ResNet18 as the surrogate model for CBS and FUS if not specified. The surrogate model is trained on the clean training set via SGD for 60 epochs, initial learning rate 0.01 and reduced by 0.1 after 30 and 50 epochs. We implement CBS according to Algorithm.1 and follow the original setting in (Xia et al., 2022) to implement FUS, i.e., 10 overall iterations and 60 epochs for updating the surrogate model in each iteration.

### 5.2 PERFORMANCE OF CBS IN TYPE I BACKDOOR ATTACKS

**Attacks & Defenses.** We consider 3 representative attacks in this category—BadNet (Gu et al., 2017) which attaches a small patch pattern as the trigger to samples to inject backdoors into neural networks; Blend (Chen et al., 2017) which applies the image blending to interpolate the trigger with samples; and Adaptive backdoor[6] (Qi et al., 2022) which introduces regularization samples to improve the stealthiness of backdoors, as backbone attacks. We include 4 representative defenses: Spectral Signiture (SS) (Tran et al., 2018), STRIP (Gao et al., 2019), Anti-Backdoor Learning (ABL) (Li et al., 2021a) and Neural Cleanser (NC) (Wang et al., 2019). We follow the default settings for backbone attacks and defenses (see Appendix A.2). For CBS, we set $\epsilon = 0.2$ and the corresponding poison rate is $0.2\%$ applied for Random and FUS. We retrain victim models on poisoned training data from scratch via SGD for 200 epochs with an initial learning rate of 0.1 and decay by 0.1 at epochs 100 and 150. Then we compare the success rate which is defined as the probability of classifying samples with triggers as the target class. We repeat every experiment 5 times and report average success rates (ASR) as well as the standard error if not specified. Results on Cifar10 are shown in Table 1 and results on Cifar100 are shown in Appendix A.4.

**Performance comparsion**. Generally, CBS enhances the resilience of backbone attacks against various defense mechanisms. It achieves notable improvement compared to Random and FUS without a significant decrease in ASR when there are no defenses in place. This is consistent with our analysis in Section 4.3. We notice that though CBS has the lowest success rate when no defenses are active, CBS it still manages to achieve commendable performance, with success rates exceeding $70\%$ and even reaching $90\%$ for certain attacks. These indicate that CBS achieves a better trade-off between stealthiness and performance. It's important to note that the effectiveness of CBS varies for different attacks and defenses. The improvements are more pronounced when dealing with stronger defenses and more vulnerable attacks. For instance, when facing SS, which is a robust defense strategy, CBS significantly enhances ASR for nearly all backbone attacks, especially for BadNet. In this case, CBS can achieve more than a $20\%$ increase compared to Random and a $15\%$ increase compared to FUS. Additionally, it's worth mentioning that the first two defense mechanisms rely

---

[5] We determine the types based on the threat models of attacking methods.

[6] Both Adaptive-Blend and Adaptive-Patch are included

Table 1: Performance on Type I backdoor attacks (Cifar10).

| Model Defense | Attacks | ResNet18 | | | ResNet18 → VGG16 | | |
|---|---|---|---|---|---|---|---|
| | | Random | FUS | CBS | Random | FUS | CBS |
| No Defenses | BadNet | 99.9±0.2 | 99.9±0.1 | 93.6±0.3 | 99.7±0.1 | 99.9±0.06 | 94.5±0.4 |
| | Blend | 89.7±1.6 | 93.1±1.4 | 86.5±0.6 | 81.6±1.3 | 86.2±0.8 | 78.3±0.6 |
| | Adapt-blend | 76.5±1.8 | 78.4±1.2 | 73.6±0.6 | 72.2±1.9 | 74.9±1.1 | 68.6±0.5 |
| | Adapt-patch | 97.5±1.2 | 98.6±0.9 | 95.1±0.8 | 93.1±1.4 | 95.2±0.7 | 91.4±0.6 |
| SS | BadNet | 0.5±0.3 | 4.7±0.2 | **20.2±0.3** | 1.9±0.9 | 3.6±0.6 | **11.8±0.4** |
| | Blend | 43.7±3.4 | 42.6±1.7 | **55.7±0.9** | 16.5±2.3 | 17.4±1.9 | **21.5±0.8** |
| | Adapt-blend | 62±2.9 | 61.5±1.4 | **70.1±0.6** | 38.2±3.1 | 36.1±1.7 | **43.2±0.9** |
| | Adapt-patch | 93.1±2.3 | 92.9±1.1 | **93.7±0.7** | 49.1±2.7 | 48.1±1.3 | **52.9±0.6** |
| STRIP | BadNet | 0.4±0.2 | 8.5±0.9 | **23.7±0.8** | 0.8±0.3 | 9.6±1.5 | **15.7±1.2** |
| | Blend | 54.7±2.7 | 57.2±1.6 | **60.6±0.9** | 49.1±2.3 | 50.6±1.7 | **56.9±0.8** |
| | Adapt-blend | 0.7±0.2 | 5.5±1.8 | **8.6±1.2** | 1.8±0.9 | 3.9±1.1 | **6.3±0.7** |
| | Adapt-patch | 21.3±2.1 | 24.6±1.8 | **29.8±1.2** | 26.5±1.7 | 27.8±1.3 | **29.7±0.5** |
| ABL | BadNet | 16.8±3.1 | 17.3±2.3 | **31.3±1.9** | 14.2±2.3 | 15.7±2.0 | **23.6±1.7** |
| | Blend | 57.2±3.8 | 55.1±2.7 | **65.7±2.1** | 55.1±1.9 | 53.8±1.3 | **56.2±1.1** |
| | Adapt-blend | 4.5±2.7 | 5.1±2.3 | **6.9±1.7** | 25.4±2.6 | 24.7±2.1 | **28.3±1.7** |
| | Adapt-patch | 5.2±2.3 | 7.4±1.5 | **8.7±1.3** | 10.8±2.7 | 11.1±1.5 | **13.9±1.3** |
| NC | BadNet | 1.1±0.7 | 13.5±0.4 | **24.6±0.3** | 2.5±0.9 | 14.4±1.3 | **17.5±0.8** |
| | Blend | 82.5±1.7 | **83.7±1.1** | 81.7±0.6 | **79.7±1.5** | 77.6±1.6 | 78.5±0.9 |
| | Adapt-blend | 72.4±2.3 | 71.5±1.8 | **74.2±1.2** | 59.8±1.7 | 59.2±1.2 | **62.1±0.6** |
| | Adapt-patch | 2.2±0.7 | 6.6±0.5 | **14.3±0.3** | 10.9±2.3 | 13.4±1.4 | **16.2±0.9** |

on detecting outliers in the latent space, and CBS consistently strengthens resistance against these defenses. This further supports the notion that boundary samples are inherently more challenging to detect and counteract. While the improvement of CBS on VGG16 is slightly less pronounced than on ResNet18, it still outperforms Random and FUS in nearly every experiment. This indicates that CBS can be effective even on unknown models.

## 5.3 Performance of CBS in Type II backdoor attacks

**Attacks & Defenses.** We consider 2 representative attacks in this category—Hidden-trigger (Saha et al., 2020) which adds imperceptible perturbations to samples to inject backdoors, and Clean-label (LC) (Turner et al., 2019) which leverages adversarial examples to train a backdoored model. We follow the default settings in the original papers, and adapt $l_2$-norm bounded perturbation (perturbation size $6/255$) for LC. We test all attacks against three representative defenses that are applicable to these attacks. We include Neural Cleanser (NC) (Wang et al., 2019), Spectral Signature (SS) (Tran et al., 2018), Fine Pruning (FP) (Liu et al., 2018), Anti-Backdoor Learning (ABL) (Li et al., 2021a). Details of these attacks and defenses are shown in Appendix A.2. We set $\epsilon = 0.3$ for CBS and $p = 0.2\%$ for Random and FUS correspondingly. For every experiment, a source class and a target class are randomly chosen, and poisoned samples are selected from the source class. The success rate is defined as the probability of misclassifying samples from the source class with triggers as the target class. Results on dataset Cifar10 and Cifar100 are presented in Table 2.

Table 2: Performance on Type II backdoor attacks.

| | Model Defense | Attacks | ResNet18 | | | ResNet18 → VGG16 | | |
|---|---|---|---|---|---|---|---|---|
| | | | Random | FUS | CBS | Random | FUS | CBS |
| CIFAR10 | No Defenses | Hidden-trigger | 81.9±1.5 | 84.2±1.2 | 76.3±0.8 | 83.4±2.1 | 86.2±1.3 | 79.6±0.7 |
| | | LC | 90.3±1.2 | 92.1±0.8 | 87.2±0.5 | 91.7±1.4 | 93.7±0.9 | 87.1±0.8 |
| | NC | Hidden-trigger | 6.3±1.4 | 5.9±1.1 | **9.7±0.9** | 10.7±2.4 | 11.2±1.5 | **14.7±0.6** |
| | | LC | 8.9±2.1 | 8.1±1.6 | **12.6±1.1** | 11.3±2.6 | 9.8±1.1 | **12.9±0.9** |
| | FP | Hidden-trigger | 11.7±2.6 | 9.9±1.3 | **14.3±0.9** | 8.6±2.4 | 8.1±1.4 | **11.8±0.8** |
| | | LC | 10.3±2.1 | 13.5±1.2 | **20.4±0.7** | 7.9±1.7 | 8.2±1.1 | **10.6±0.7** |
| | ABL | Hidden-trigger | 1.7±0.8 | 5.6±1.6 | **10.5±1.1** | 3.6±1.1 | 8.8±0.8 | **10.4±0.6** |
| | | LC | 0.8±0.3 | 8.9±1.5 | **12.1±0.8** | 1.5±0.7 | 9.3±1.2 | **12.6±0.8** |
| CIFAR100 | No Defenses | Hidden-trigger | 80.6±2.1 | 84.1±1.8 | 78.9±1.3 | 78.2±2.3 | 81.4±1.6 | 75.8±1.2 |
| | | LC | 86.3±2.3 | 87.2±1.4 | 84.7±0.9 | 84.7±2.8 | 85.2±1.4 | 81.5±1.1 |
| | NC | Hidden-trigger | 3.8±1.4 | 4.2±0.9 | **7.6±0.7** | 4.4±1.1 | 5.1±1.2 | **6.8±0.9** |
| | | LC | 6.1±1.8 | 5.4±1.1 | **8.3±0.5** | 3.9±1.2 | 3.8±0.9 | **8.3±0.7** |
| | FP | Hidden-trigger | 15.3±3.1 | 16.7±0.9 | **23.2±0.7** | 8.9±1.3 | 9.3±1.1 | **12.3±0.7** |
| | | LC | 13.8±2.7 | 12.7±1.5 | **16.9±0.6** | 10.3±1.4 | 9.9±0.8 | **14.2±0.5** |
| | ABL | Hidden-trigger | 2.3±0.9 | 3.9±1.3 | **6.5±1.1** | 3.7±0.9 | 3.5±0.7 | **6.4±0.4** |
| | | LC | 0.9±0.2 | 2.7±0.8 | **6.2±0.6** | 2.5±0.8 | 2.1±0.7 | **6.7±0.5** |

**Performance comparison.** As presented in Table 2, our method CBS, demonstrates similar behavior to Type I attacks, showing enhanced resistance against various defense mechanisms at the cost of some success rate. Notably, CBS consistently outperforms Random and FUS when defenses are in place, highlighting its versatility in different scenarios. Particularly for vulnerable attacks like Bad-Net, CBS achieves substantial improvements, surpassing Random by over $10\%$ and FUS by over $5\%$. Furthermore, CBS exhibits smaller standard errors, indicating its higher stability. However, there is still room for further improvement, as the absolute success rate is not as impressive as in Type I attacks. We consider this area for future research endeavors.

### 5.4 PERFORMANCE OF CBS IN TYPE III BACKDOOR ATTACKS

**Attacks & Defenses.** We consider 3 Representative attacks in this category—Lira (Doan et al., 2021b) which involves a stealthy backdoor transformation function and iteratively updates triggers and model parameters; WaNet (Nguyen & Tran, 2021) which applies the image warping technique to make triggers more stealthy; Wasserstein Backdoor (WB) (Doan et al., 2021a) which directly minimizes the distance between poisoned and clean representations. Note that Type III attacks allow the attackers to take control of the training process. Though our threat model does not require this additional capability of attackers, we follow this assumption when implementing these attacks. Therefore, we directly select samples based on ResNet18 and VGG16 rather than using ResNet18 as a surrogate model. We conduct 3 representative defenses that are applicable for this type of attacks—Neural Cleanser (NC) (Wang et al., 2019), STRIP (Gao et al., 2019), Fine Pruning (FP) (Liu et al., 2018). We follow the default settings to implement these attacks and defenses (details in Appendix A.2). We set $\epsilon = 0.37$ which matches the poison rate $p = 0.1$ in the original settings of backbone attacks. Results on Cifar10 and Cifar100 are presented in Table 3.

**Performance comparison.** Except for the common findings in previous attacks, where CBS consistently outperforms baseline methods in nearly all experiment, we observe that the impact of CBS varies when applied to different backbone attacks. Specifically, CBS tends to yield the most significant improvements when applied to WB, while its effect is less pronounced when applied to WaNet. For example, when confronting FP and comparing CBS with both Random and FUS, we observed an increase in ASR of over $7\%$ on WB, while the increase on WaNet amounted to only $3\%$, with Lira showing intermediate results. This divergence may be attributed to the distinct techniques employed by these attacks to enhance their resistance against defenses. WB focuses on minimizing the distance between poisoned samples and clean samples from the target class in the latent space. By selecting boundary samples that are closer to the target class, WB can reach a smaller loss than that optimized on random samples, resulting in improved resistance. The utilization of the fine-tuning process and additional information from victim models in Lira enable a more precise estimation of decision boundaries and the identification of boundary samples. WaNet introduces Gaussian noise to some randomly selected trigger samples throughout the poisoned dataset, which may destroy the impact of CBS if some boundary samples move away from the boundary after adding noise. These observations suggest that combining CBS with proper trigger designs can achieve even better performance, and it is an interesting topic to optimize trigger designs and sampling methods at the same time for more stealthiness, which leaves for future exploration.

### 5.5 ABLATION STUDY

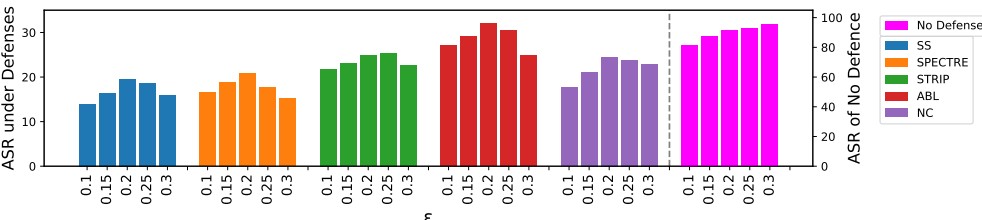

Figure 4: An illustration on the influence of $\epsilon$ in CBS when applied to BadNet. The magenta bar represents ASR without defenses while the left bars present ASR under defenses.

**Impact of $\epsilon$.** Threshold $\epsilon$ is one key hyperparameter in CBS to determine which samples are around the boundary, and to study the impact of $\epsilon$, we conduct experiments on different $\epsilon$. Since the size of the poisoned set generated by different $\epsilon$ is different, we fix the poison rate to be $0.1\%$ (50 samples), and for large $\epsilon$ that generates more samples, we randomly choose 50 samples from it to form the final poisoned set. We consider $\epsilon = 0.1, 0.15, 0.2, 0.25, 0.3$, and conduct experiments on model

Table 3: Performance on Type III backdoor attacks.

| | Model Defense | Attacks | ResNet18 Random | ResNet18 FUS | ResNet18 CBS | VGG16 Random | VGG16 FUS | VGG16 CBS |
|---|---|---|---|---|---|---|---|---|
| CIFAR10 | No Defenses | Lira | 91.5±1.4 | 92.9±0.7 | 88.2±0.8 | 98.3±0.8 | 99.2±0.5 | 93.6±0.4 |
| | | WaNet | 90.3±1.6 | 91.4±1.3 | 87.9±0.7 | 96.7±1.4 | 97.3±0.9 | 94.5±0.5 |
| | | WB | 88.5±2.1 | 90.9±1.9 | 86.3±1.2 | 94.1±1.1 | 95.7±0.8 | 92.8±0.7 |
| | NC | Lira | 10.3±1.6 | 12.5±1.1 | **16.1±0.7** | 14.9±1.5 | 18.3±1.1 | **19.6±0.8** |
| | | WaNet | 8.9±1.5 | 10.1±1.3 | **13.4±0.9** | 10.5±1.1 | 12.2±0.7 | **13.7±0.9** |
| | | WB | 20.7±2.1 | 19.6±1.2 | **27.2±0.6** | 23.1±1.3 | 24.9±0.8 | **28.7±0.5** |
| | STRIP | Lira | 81.5±3.2 | 82.3±2.3 | **87.7±1.1** | 82.8±2.4 | 81.5±1.7 | **84.6±1.3** |
| | | WaNet | 80.2±3.4 | 79.7±2.5 | **86.5±1.4** | 77.6±3.1 | **79.3±2.2** | 78.2±1.5 |
| | | WB | 80.1±2.9 | 81.7±1.8 | **86.6±1.2** | 83.4±2.7 | 82.6±1.8 | **87.3±1.1** |
| | FP | Lira | 6.7±1.7 | 6.2±1.2 | **12.5±0.7** | 10.4±1.1 | 9.8±0.8 | **13.3±0.6** |
| | | WaNet | 4.8±1.3 | 6.1±0.9 | **8.2±0.8** | 6.8±0.9 | 6.4±0.6 | **8.3±0.4** |
| | | WB | 20.8±2.3 | 21.9±1.7 | **28.3±1.1** | 25.7±1.3 | 26.2±1.2 | **29.1±0.7** |
| CIFAR100 | No Defenses | Lira | 98.2±0.7 | 99.3±0.2 | 96.1±1.3 | 97.1±0.8 | 99.3±0.4 | 94.5±0.5 |
| | | WaNet | 97.7±0.9 | 99.1±0.4 | 94.3±1.2 | 96.3±1.2 | 98.7±0.9 | 94.1±0.7 |
| | | WB | 95.1±0.6 | 96.4±1.1 | 94.7±0.9 | 93.2±0.9 | 96.7±0.4 | 91.9±0.8 |
| | NC | Lira | 0.2±0.1 | 1.7±1.2 | **5.8±0.9** | 3.4±0.7 | 3.9±1.0 | **7.2±0.9** |
| | | WaNet | 1.6±0.8 | 3.4±1.3 | **8.2±0.8** | 2.9±0.6 | 2.5±0.8 | **5.1±1.2** |
| | | WB | 7.7±1.5 | 7.5±0.9 | **15.7±0.7** | 8.5±1.3 | 7.6±0.9 | **14.9±0.7** |
| | STRIP | Lira | 84.3±2.7 | 83.7±1.5 | **87.2±1.1** | 82.7±2.5 | 83.4±1.8 | **87.8±1.4** |
| | | WaNet | 82.5±2.4 | 82.0±1.6 | **83.9±0.9** | 81.4±2.7 | **84.5±1.7** | 82.6±0.8 |
| | | WB | 85.8±1.9 | 86.4±1.2 | **88.1±0.8** | 82.9±2.4 | 82.3±1.5 | **86.5±1.4** |
| | FP | **Lira** | 7.4±1.9 | 8.9±1.1 | **15.2±0.9** | 8.5±3.2 | 11.8±2.4 | **14.7±1.1** |
| | | **WaNet** | 6.7±1.7 | 6.3±0.9 | **11.3±0.7** | 9.7±2.9 | 9.3±1.8 | **12.6±1.3** |
| | | **WB** | 19.2±1.5 | 19.7±0.7 | **26.1±0.5** | 17.6±2.4 | 18.3±1.7 | **24.9±0.8** |

ResNet18 and dataset Cifar10 with BadNet as the backbone. Results of ASR under no defense and 5 defenses are shown in Figure 4. It is obvious that the ASR for no defenses is increasing when $\epsilon$ is increasing. We notice that large $\epsilon$ (0.25,0.3) has higher ASR without defenses but relatively small ASR against defenses, indicating that the stealthiness of backdoors is reduced for larger $\epsilon$. For small $\epsilon$ (0.1), ASR decreases for either no defenses or against defenses. These observations suggest that samples too close or too far from the boundary can hurt the effect of CBS, and a proper $\epsilon$ is needed to balance between performance and stealthiness.

**Impact of confidence.** Since our core idea is to select samples with lower confidence, we conduct experiments to compare the influence of high-confidence and low-confidence samples. In detail, we select low-confidence samples with $\epsilon = 0.2$ and high-confidence samples with $\epsilon = 0.9$[7]. We still conduct experiments on ResNet18 and Cifar10 with Bad-Net, and the ASR is shown in Figure 5. Note that low-confidence samples significantly outperform the other 2 types of samples, while high-confidence samples are even worse than random samples. Therefore, these results further support our claim that low-confidence samples can improve the stealthiness of backdoors.

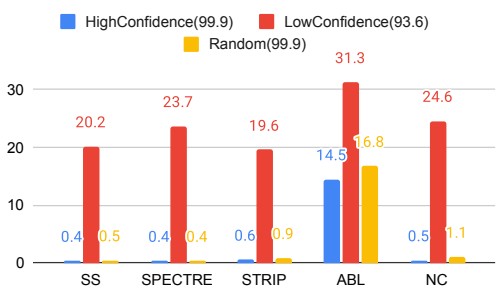

Figure 5: Illustrating impacts of confidence.

## 6 CONCLUSION

In this paper, we highlight a crucial aspect of backdoor attacks that was previously overlooked. We find that the choice of which samples to poison plays a significant role in a model's ability to resist defense mechanisms. To address this, we introduce a confidence-driven boundary sampling approach, which involves carefully selecting samples near the decision boundary. This approach has proven highly effective in improving an attacker's resistance against defenses. It also holds promising potential for enhancing the robustness of all backdoored models against defense mechanisms.

---

[7]Here we refer to the different direction of Eq.4.1, i.e. $|s_c(f(x;\theta))_y - s_c(f(x;\theta))_{y'}| \geq \epsilon$

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

# A   APPENDIX

## A.1   PROOF OF THEOREM 4.1

Recall the settings in Section 4.3. Suppose two classes $C_1, C_2$ form two uniform distributions of balls centered at $\mu_1, \mu_2$ with radius $r$ in the latent space, i.e.

$$C_1 \sim p_1(x) = \frac{1}{\pi r^2} 1[\|x - \mu_1\|_2 \leq r], \text{ and } C_2 \sim p_2(x) = \frac{1}{\pi r^2} 1[\|x - \mu_2\|_2 \leq r]$$

Both classes have $n$ samples. Assume $x \in C_1$, and a trigger is added to $x$ such that $\tilde{x} = x + \epsilon t / \|t\|_2 := x + a$. Then define the poisoned data as $\tilde{C}_1 = C_1/\{x\}$ and $\tilde{C}_2 = C_1 \cup \{\tilde{x}\}$. Then we train a backdoored SVM on the poisoned data. The following theorem provides estimations for Mahalanobis distance which serves as the indicator of outliers, and success rate.

**Theorem A.1.** *Assume $\tilde{x} = x + \epsilon t / \|t\|_2 := x + a$ for some trigger $t$ and strength $\epsilon$, and $(\mu_2 - \mu_1)^T \geq 0$. Mahalanobis distance between the poisoned sample $\tilde{x}$ and the target class $\tilde{C}_2$ is*

$$d_M^2(\tilde{x}, \tilde{C}_2) = \frac{4n}{(n+1)r^2} \frac{(n+1)r^2}{(n+1)r^2 / \|\tilde{x}\|_2^2 + 4}.$$

*In addition, the success rate is an increasing function of*

$$\epsilon \cos(a, \tilde{x} - \mu_1) - \|\tilde{x} - \mu_1\|/2 - r/2$$

*Proof.* We use Figure 6 as illustrations and help proof. Given class 1 and class 2, the decision boundary for SVM is approximately $(\mu_2 - \mu_1)^T(x - \frac{\mu_1 + \mu_2}{2}) = 0$, i.e. $2(\mu_2 - \mu_1)^T x = \|\mu_2\|_2^2 - \|\mu_1\|_2^2$.

Consider the following backdoor attack: select one sample from $C_1$ to add a trigger $t$ and force it moving towards $C_2$. In this case we have the poisoned sample $\tilde{x} = x + \epsilon \frac{t}{\|t\|}$, where $(\mu_2 - \mu_1)^T t \geq 0$ to ensure the poisoned sample is moving towards target class $C_2$. We further label $\tilde{x}$ as class 2 and obtain poisoned training $\tilde{C}_1 = C_1/\{\tilde{x}\}$ and $\tilde{C}_2 = C_2 \cup \{\tilde{x}\}$. Denote $\hat{\mu}_1$ and $\hat{\mu}_2$ as the mean of the clean samples from $C_1$ and $C_2$ respectively, and let $\hat{\mu}_2 = 0$ for simplicity. Then we have the mean $\tilde{\mu}_2$ and covariance matrix $\tilde{\Sigma}_2$ for class $\tilde{C}_2$ as follows:

$$\tilde{\mu}_2 = \mathbb{E}_{x \sim \tilde{C}_2} x = \frac{n}{n+1} \mathbb{E}_{x \sim C_2} x + \frac{1}{n+1} \tilde{x} = \frac{1}{n+1} \tilde{x}$$

$$\tilde{\Sigma}_2 = \mathbb{E}_{x \sim \tilde{C}_2} x x^T - \tilde{\mu}_2 \tilde{\mu}_2^T$$

$$= \mathbb{E}1[x \in C_2] x x^T + \mathbb{E}1[x \text{ is poisoned}] x x^T - \tilde{\mu}_2 \tilde{\mu}_2^T$$

$$= \frac{n}{n+1} \mathbb{E}_{x \in C_2} x x^T + \frac{1}{n+1} \tilde{x} \tilde{x}^T - \tilde{\mu}_2 \tilde{\mu}_2^T$$

$$= \frac{n}{n+1} \frac{r^2}{4} I + \frac{1}{n+1} \tilde{x} \tilde{x}^T - \tilde{\mu}_2 \tilde{\mu}_2^T$$

We can compute the Mahalanobis distance between $\tilde{x}$ and $\tilde{C}_2$ as $d_M^2(\tilde{x}, \tilde{D}_2) = (\tilde{x} - \tilde{\mu}_2)^T \tilde{\Sigma}_2^{-1}(\tilde{x} - \tilde{\mu}_2)$. Then we have:

$$
\begin{aligned}
d_M^2(\tilde{x}, \tilde{C}_{2,x}) &= (\tilde{x} - \tilde{\mu}_2)^T \tilde{\Sigma}_2^{-1} (\tilde{x} - \tilde{\mu}_2) \\
&= \left( \tilde{x} - \frac{1}{n+1}\tilde{x} \right)^T \left[ \frac{n}{n+1} \frac{r^2}{4} I_d + \frac{n}{(n+1)^2} \tilde{x}\tilde{x}^T \right]^{-1} \left( \tilde{x} - \frac{1}{n+1}\tilde{x} \right) \quad (5) \\
&= \frac{4(n+1)}{nr^2} \left( \frac{n}{n+1} \right)^2 \tilde{x}^T \left[ I - \frac{\frac{4}{(n+1)r^2}\tilde{x}\tilde{x}^T}{1 + \frac{4}{(n+1)r^2}\tilde{x}^T\tilde{x}} \right] \tilde{x} \quad (6) \\
&= \frac{4n}{(n+1)r^2} \left[ \tilde{x}^T\tilde{x} - \frac{\frac{4}{(n+1)r^2}\tilde{x}^T\tilde{x}\tilde{x}^T\tilde{x}}{1 + \frac{4}{(n+1)r^2}\tilde{x}^T\tilde{x}} \right] \\
&= \frac{4n}{(n+1)r^2} \left[ \|\tilde{x}\|_2^2 - \frac{4\|\tilde{x}\|_2^4}{(n+1)r^2 + 4\|\tilde{x}\|_2^2} \right] \\
&= \frac{4n}{(n+1)r^2} \frac{(n+1)r^2\|\tilde{x}\|_2^2 + 4\|\tilde{x}\|_2^4 - 4\|\tilde{x}\|_2^4}{(n+1)r^2 + 4\|\tilde{x}\|_2^2} \\
&= \frac{4n}{(n+1)r^2} \frac{(n+1)r^2}{(n+1)r^2/\|\tilde{x}\|_2^2 + 4} \quad (7)
\end{aligned}
$$

where from Eq. 5 to Eq. 6 we use Sherman-Morrison equation. It is obvious that the final result in Eq.7 is monotonically increasing with respect to $\|\tilde{x}\|_2^2$. If we consider two selections of $x$: (1) select $x_b$ such that $\|x_b - \hat{\mu}_2\|_2 = \min_{x \in C_1} \|x - \hat{\mu}_2\|_2$, i.e. the point closest to the decision boundary, which corresponds to CBS; (2) randomly select $x_u \in C_1$ following the uniform distribution, which aligns with the sampling scheme in existing literature. Then we immediately have $\|\tilde{x}_b\|_2^2 \leq \|\tilde{x}_u\|_2^2$, and therefore $d_M^2(\tilde{x}_b, \tilde{C}_2) \leq d_M^2(\tilde{x}_u, \tilde{C}_2)$. This implies that samples from CBS is harder to detect.

Next, let us take a look at decision boundaries derived from different poisoning samples. We assume $n \to \infty$ for simplicity. As shown in Fig.6a (for Random) and 6b (for CBS), for a given sample $x$ (red point), $\tilde{x} = x + \epsilon\frac{t}{\|t\|_2} := x + a$ (blue point), where $\mu_1^T t \geq 0$. Since $C_2$ is not changed, the backdoored decision boundary (the bold black line) is determined by $\tilde{x}$ and $C_1$. Specifically, the decision boundary is determined by $\tilde{x}$ and center $\mu_1$. Connect the center of $C_1$ with $\tilde{x}$ and we obtain an interaction point on $C_1$, which is $\mu_1 + r\frac{\tilde{x}-\mu_1}{\|\tilde{x}-\mu_1\|_2}$ and the center between it and $\tilde{x}$ is

$$
\tilde{c}_1 = \frac{\mu_1 + r\frac{\tilde{x}-\mu_1}{\|\tilde{x}-\mu_1\|_2} + \tilde{x}}{2}. \quad (8)
$$

Then we can derive the equation for the backdoored decision boundary:

$$
(x - \tilde{c}_1)^T(\tilde{x} - \mu_1) = 0 \quad (9)
$$

where we assume this decision boundary is not overlapped with $C_2$. During the inference, triggers will be added to samples in $C_1$, which means that the circle of $C_1$ will shift by $\epsilon\frac{t}{\|t\|_2}$ (denoted as $\bar{C}_1$) as shown in Fig.6a and 6b, then the yellow area will be misclassified as $C_2$. Thus the success rate without any defenses is determined by the area of the yellow area. Since the circle of $C_1$ is fixed, we only need to compare the distance from the center of $\bar{C}_1$ to the backdoored decision boundary, which is the bold green line in Fig.6a and 6b. Notice that $\mu_1 - \tilde{x}$ is orthogonal to the decision boundary defined in Eq.9, thus the length of the green bold line is the length of $\tilde{\tilde{c}}_1 - \tilde{c}_1$ in the direction of $\mu_1 - \tilde{x}$ where $\tilde{\tilde{c}}_1$ is the center of $\tilde{\bar{C}}_1$, thus the distance is computed as:

$$
\begin{aligned}
d_D(\tilde{x}) &= \frac{(\tilde{\tilde{c}}_1 - \tilde{c}_1)^T(\mu_1 - \tilde{x})}{\|\mu_1 - \tilde{x}\|_2} = \frac{1}{\|\tilde{x} - \mu_1\|} \left( \mu_1 + a - \frac{\mu_1 + r\frac{\tilde{x}-\mu_1}{\|\tilde{x}-\mu_1\|} + \tilde{x}}{2} \right)^T (\tilde{x} - \mu_1) \\
&= \frac{a^T(\tilde{x} - \mu_1)}{\|\tilde{x} - \mu_1\|} - \frac{\|\tilde{x} - \mu_1\|}{2} - \frac{r}{2} \quad (10) \\
&= \|a\|\cos(a, \tilde{x} - \mu_1) - \frac{\|\tilde{x} - \mu_1\|}{2} - \frac{r}{2}
\end{aligned}
$$

This finishes the proof. This formulation indicates that smaller $\|\tilde{x} - \mu_1\|$ and $\cos(a, \tilde{x} - \mu_1)$ leads to larger $d_D(\tilde{x})$. Therefore, the closer the selected sample to the decision boundary, the smaller the

area of the yellow area is, and the smaller the success rate for the backdoored attack. Note that here we only consider the case that $a^T(\tilde{x} - \mu_1) \geq 0$ otherwise the poisoned sample will remain in the original $C_1$. These results reveal the trade-off between stealthiness and performance.

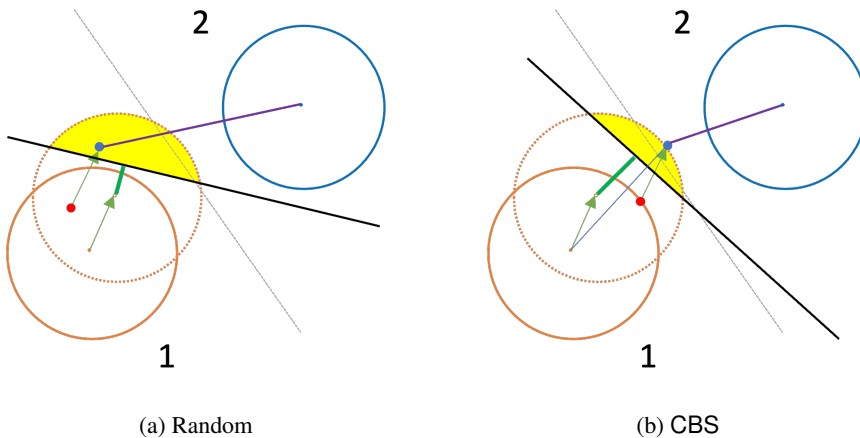

(a) Random           (b) CBS

Figure 6: Illustrating figures for SVM under Random and CBS. The red point is a sample $x$ from $C_1$, and the blue one is the triggered sample $\tilde{x}$. The grey dashed line and black bold line represent the decision boundary of clean and backdoored SVM respectively. We are interested in the Mahalanobis distance between $\tilde{x}$ and the target class $\tilde{C}_2$. The yellow area is in proportion to the success rate and the length of the green bold line is positively correlated with the area of the yellow part. It is obvious that CBShas smaller Mahalanobis distance and smaller area of yellow.

□

## A.2   IMPLEMENTATION DETAILS

In this section, we provide details of attacks and defenses used in experiments as well as implementation details[8].

### A.2.1   IMPLEMENTATIONS FOR SAMPLINGS

We implement Random with a uniform distribution on $\mathcal{D}_{tr}$. We implement CBS according to Algorithm 1, and the surrogate model is trained via SGD for 60 epochs with an initial learning rate of 0.01 and decreases by 0.1 at epochs 30,50. We implement FUS according to its original settings, i.e. 10 overall iterations and 60 epochs for updating surrogate model in each iteration, and the surrogate model is pretrained same as in CBS.

### A.2.2   ATTACKS

We will provide brief introduction and implementation details for all the backbone attacks implemented in this work.

**BadNet (Gu et al., 2017)**. BadNet is the first work exploring the backdoor attacks, and it attaches a small patch to the sample to create the poisoned training set. Then this training set is used to train a backdoor model. We implement it based on the code of work (Qi et al., 2022) and following the default setting.

**Blend (Chen et al., 2017)**. Blend incorporates the image blending technique, and blends the selected image with a pre-specified trigger pattern that has the same size as the original image. We implement this attack based on the code of work (Qi et al., 2022), and following the default setting, i.e. mixing ratio $\alpha = 0.2$.

---

[8]Code can be found at https://anonymous.4open.science/r/boundary-backdoor-E313

**Adaptive backdoor (Qi et al., 2022)**. This method leverages regularization samples to weaken the relationship between triggers and the target label and achieve better stealthiness. We implement two versions of the method: Adaptive-blend and Adaptive-patch. During the implementation, we consider the conservatism ratio of $\eta = 0.5$ and mixing ratio $\alpha = 0.2$ for adaptive-blend; conservatism ratio $\eta = 2/3$ and 4 patches for Adaptive-patch.

**Hidden-trigger (Saha et al., 2020)**. This attacking method first attaches the trigger to a sample and then searches for an imperceptible perturbation that achieves a similar model output (measured by $l_2$ norm) as the triggered sample. We follow the original settings in work (Saha et al., 2020), i.e. placing the trigger at the right corner of the image, setting the budget size as $16/255$, optimizing the perturbation for 10000 iterations with a learning rate of 0.01 and decay by 0.95 for every 2000 iterations.

**Label-consistent (LC) (Turner et al., 2019)**. This attacking method leverages GAN or adversarial examples to create the poisoned image without changing the label. We implement the one with adversarial examples bounded by $l_2$ norm. We set the budget size as $600$ to achieve a higher success rate.

**Lira (Doan et al., 2021b)**. This method iteratively learns the model parameters and a trigger generator. Once the trigger generator is trained, attackers will finetune the model on poisoned samples attached with triggers generated by the generator, and release the backdoored model to the public. Our implementation is based on the Benchmark (Wu et al., 2022).

**WaNet (Nguyen & Tran, 2021)**. WaNet incorporates the image warping technique to inject invisible triggers into the selected image. To improve the poisoning effect, they introduce a special training mode that add Gaussian noise to the warping field to improve the success rate. Our implementation is based on the Benchmark (Wu et al., 2022).

**Wasserstein Backdoor (WB) (Doan et al., 2021a)**. This method directly minimizes the distance between poisoned samples and clean samples in the latent space. We follow the original settings, i.e. training 50 epochs for Stage I and 450 epochs for Stage II, set the threshold of constraint as 0.01.

### A.2.3 Defenses

**Spectral Signature (SS) (Tran et al., 2018)**. This defense detects poisoned samples with stronger spectral signatures in the learned representations. We remove $1.5 * p$ of samples in each class.

**Activation Clustering (AC) (Chen et al., 2018)**. This defense is based on the clustering of activations of the last hidden neural network layer, for which clean samples and poisoned samples form distinct clusters. We remove clusters with sizes smaller than $35\%$ for each class.

**SCAn (Tang et al., 2021)**. This defense leverages an EM algorithm to decompose an image into its identity part and variation part, and a detection score is constructed by analyzing the distribution of the variation.

**SPECTRE (Hayase et al., 2021)**. This method proposes a novel defense algorithm using robust covariance estimation to amplify the spectral signature of corrupted data. We also remove $1.5 * p$ of samples in each class.

**Fine Pruning (FP) (Liu et al., 2018)**. This is a model-pruning-based backdoor defense that eliminates a model's backdoor by pruning these dormant neurons until a certain clean accuracy drops.

**STRIP (Gao et al., 2019)**. STRIP is a sanitation-based method relying on the observation that poisoned samples are easier to be perturbed, and detect poisoned samples through adversarial perturbations.

**Neural Cleanse (NC) (Wang et al., 2019)**. This is a trigger-inversion method that restores triggers by optimizing the input domain. It is based on the intuition that the norm of reversed triggers from poisoned samples will be much smaller than clean samples.

**Anti-Backdoor Learning (ABL) (Li et al., 2021a)**. This defense utilizes local gradient ascent to isolate $1\%$ suspected training samples with the smallest losses and leverage unlearning techniques to train a cleansed model on poisoned data.

A.3 ALGORITHMS

In this section, we provide detailed algorithms for CBS and its application on Blend (Chen et al., 2017).

As shown in Algorithm 1, CBS first pretrain a surrogate model $f(\cdot; \theta)$ on the clean training set $\mathcal{D}_{tr}$ for $E$ epochs; then $f(\cdot; \theta)$ is used to estimate the confidence score for every sample; for a given target $y^t$, samples satisfying $|s_c(f(x_i; \theta))_{y_i} - s_c(f(x_i; \theta))_{y^t}| \le \epsilon$ are selected as the poison sample set $U$.

As shown in Algorithm 2, the poison sample set $U$ is first selected via Algorithm 1; then for each sample in $U$, a trigger is blended to this sample with a mixing ratio $\alpha$ via $x' = \alpha * t + (1 - \alpha) * x$ and generate the poisoned training set $D_p$.

---

**Algorithm 1** CBS

**Input** Clean training set $\mathcal{D}_{tr} = \{(x_i, y_i)\}_{i=1}^N$, model $f(\cdot; \theta)$, pre-train epochs $E$, threshold $\epsilon$, target class $y^t$
**Output** Poisoned sample set $U$, poisoned label set $S_p$.
    Pre-train the surrogate model $f$ on $\mathcal{D}_{tr}$ for $T$ epochs and obtain $f(\cdot; \theta)$
    Initialize poisoned sample set $U = \{\}$
    **for** $i = 1, ..., N$ **do**
        **if** $|s_c(f(x_i; \theta))_{y_i} - s_c(f(x_i; \theta))_{y^t}| \le \epsilon$ **then**
            $U = U \cup \{(x_i, y_i)\}$
        **else** Continue
        **end if**
    **end for**
    Return poisoned sample set $U$

---

**Algorithm 2** Blend+CBS

**Input** Clean training set $\mathcal{D}_{tr} = \{(x_i, y_i)\}_{i=1}^N$, surrogate model $f(\cdot; \theta)$, pre-train epochs $E$, threshold $\epsilon$, target class $y^t$, mixing ratio $\alpha$, trigger pattern $t$
**Output** Poisoned training set $D_p$
    Initialize poisoned training set $D_p$
    Select poison set $U$ from $\mathcal{D}_t r$ via Algorithm.1
    **for** $x \in U$ **do**
        Inject triggers to samples: $x' = \alpha * t + (1 - \alpha) * x$
        $D_p = D_p \cup \{x'\}$
    **end for**
    Return poisoned training set $D_p$

---

A.4 ADDITIONAL EXPERIMENTS

In this section, we provide additional experimental results.

**Type I attacks.** We include additional defenses: Activation Clustering (AC) (Chen et al., 2018), SCAn (Tang et al., 2021), SPECTRE (Hayase et al., 2021), Fine Pruning (FP) (Liu et al., 2018). We also conduct experiments on Cifar100. Results of Type I attacks on Cifar10 and Cifar100 datasets are shown in Table 4 and 5 respectively. CBS has similar behavior on Cifar100—improve the resistance against various defenses while slightly decrease ASR without defenses.

**Type II attacks.** We also include additional defenses: Spectral Signature (SS) (Tran et al., 2018). The results of all defenses on model ResNet18, VGG16 and datasets Cifar10, Cifar100 are presented in Table 6. Detailed analysis is shown in Section 5.3.

**Type III attacks.** For a more comprehensive evaluations, we include additional experiments: Activation Clustering (AC) (Chen et al., 2018) and Spectral Signature (SS) (Tran et al., 2018). All results including model ResNet18, VGG16 and datasets Cifar10, Cifar100 are shown in Table 7. Detailed analysis can be found in Section 5.4.

Table 4: Full Performance on Type I backdoor attacks (Cifar10).

| Model Defense | Attacks | ResNet18 | | | ResNet18 → VGG16 | | |
|---|---|---|---|---|---|---|---|
| | | **Random** | **FUS** | **CBS** | **Random** | **FUS** | **CBS** |
| No Defenses | **BadNet** | 99.9±0.2 | 99.9±0.1 | 93.6±0.3 | 99.7±0.1 | 99.9±0.06 | 94.5±0.4 |
| | **Blend** | 89.7±1.6 | 93.1±1.4 | 86.5±0.6 | 81.6±1.3 | 86.2±0.8 | 78.3±0.6 |
| | **Adapt-blend** | 76.5±1.8 | 78.4±1.2 | 73.6±0.6 | 72.2±1.9 | 74.9±1.1 | 68.6±0.5 |
| | **Adapt-patch** | 97.5±1.2 | 98.6±0.9 | 95.1±0.8 | 93.1±1.4 | 95.2±0.7 | 91.4±0.6 |
| SS | **BadNet** | 0.5±0.3 | 4.7±0.2 | **23.2±0.3** | 1.9±0.9 | 3.6±0.6 | **11.8±0.4** |
| | **Blend** | 43.7±3.4 | 42.6±1.7 | **55.7±0.9** | 16.5±2.3 | 17.4±1.9 | **21.5±0.8** |
| | **Adapt-blend** | 62±2.9 | 61.5±1.4 | **70.1±0.6** | 38.2±3.1 | 36.1±1.7 | **43.2±0.9** |
| | **Adapt-patch** | 93.1±2.3 | 92.9±1.1 | **93.7±0.7** | 49.1±2.7 | 48.1±1.3 | **52.9±0.6** |
| AC | **BadNet** | 0.6±0.3 | 14.2±0.9 | **20.5±0.7** | 5.7±1.2 | 5.3±1.3 | **10.5±1.5** |
| | **Blend** | 77.1±2.8 | **79.6±2.6** | 77.8±1.4 | 83.1±3.5 | **83.2±2.4** | 81.4±2.1 |
| | **Adapt-blend** | 76.8±2.1 | 76.1±1.4 | **79.3±1.6** | 69.9±2.8 | 70.6±1.5 | **73.1±1.2** |
| | **Adapt-patch** | **97.5±2.6** | 94.2±1.7 | 96.6±0.9 | 92.4±2.7 | **93.2±1.4** | 91.3±1.3 |
| SCAn | **BadNet** | 0.7±0.4 | 10.7±1.2 | **23.5±0.8** | 12.4±1.5 | 10.7±1.2 | **26.4±1.1** |
| | **Blend** | **84.4±3.4** | 83.6±2.5 | 78.3±2.6 | 80.6±3.2 | **82.1±2.4** | 78.2±0.9 |
| | **Adapt-blend** | 78.2±2.6 | 77.5±2.1 | **81.5±1.4** | 71.9±2.5 | 71.1±2.1 | **74.4±1.3** |
| | **Adapt-patch** | **97.5±0.9** | 94.1±0.8 | 96.9±0.4 | 93.1±1.1 | **93.8±0.9** | 91.5±0.5 |
| STRIP | **BadNet** | 0.4±0.2 | 8.5±0.9 | **26.2±0.8** | 0.8±0.3 | 9.6±1.5 | **15.7±1.2** |
| | **Blend** | 54.7±2.7 | 57.2±1.6 | **60.6±0.9** | 49.1±2.3 | 50.6±1.7 | **56.9±0.8** |
| | **Adapt-blend** | 0.7±0.2 | 5.5±1.8 | **8.6±1.2** | 1.8±0.9 | 3.9±1.1 | **6.3±0.7** |
| | **Adapt-patch** | 21.3±2.1 | 24.6±1.8 | **29.8±1.2** | 26.5±1.7 | 27.8±1.3 | **29.7±0.5** |
| SPECTRE | **BadNet** | 0.9±0.5 | 10.1±1.4 | **19.6±1.3** | 0.7±0.3 | 8.7±1.2 | **14.9±0.8** |
| | **Blend** | 9.2±2.4 | 16.7±2.1 | **24.2±1.7** | 8.7±2.6 | 12.8±1.9 | **18.6±0.9** |
| | **Adapt-blend** | 69±3.5 | 66.8±2.7 | **70.3±1.8** | 67.9±3.2 | 65.2±1.8 | **69.4±0.9** |
| | **Adapt-patch** | 91.4±1.4 | 89.4±1.2 | **93.1±0.7** | 92.5±2.4 | 91.8±1.4 | **92.1±1.2** |
| ABL | **BadNet** | 16.8±3.1 | 17.3±2.3 | **31.3±1.9** | 14.2±2.3 | 15.7±2.0 | **23.6±1.7** |
| | **Blend** | 57.2±3.8 | 55.1±2.7 | **65.7±2.1** | 55.1±1.9 | 53.8±1.3 | **56.2±1.1** |
| | **Adapt-blend** | 4.5±2.7 | 5.1±2.3 | **6.9±1.7** | 25.4±2.6 | 24.7±2.1 | **28.3±1.7** |
| | **Adapt-patch** | 5.2±2.3 | 7.4±1.5 | **8.7±1.3** | 10.8±2.7 | 11.1±1.5 | **13.9±1.3** |
| FP | **BadNet** | 75.2±3.2 | 80.8±2.4 | **81.2±1.3** | 68.3±3.1 | 70.5±2.3 | **73.7±1.1** |
| | **Blend** | 79.5±3.7 | **81.5±2.4** | 80.4±1.5 | 70.2±2.9 | 72.5±2.1 | **79.3±1.5** |
| | **Adapt-blend** | **77.5±2.7** | 75.3±2.3 | 77.4±1.2 | 65.1±3.4 | 64.2±2.7 | **68.5±1.6** |
| | **Adapt-patch** | **97.5±1.1** | 92.7±2.3 | 96.3±0.9 | 93.4±2.2 | 93.3±1.7 | **93.7±0.8** |
| NC | **BadNet** | 1.1±0.7 | 13.5±0.4 | **24.6±0.3** | 2.5±0.9 | 14.4±1.3 | **17.5±0.8** |
| | **Blend** | 82.5±1.7 | **83.7±1.1** | 81.7±0.6 | **79.7±1.5** | 77.6±1.6 | 78.5±0.9 |
| | **Adapt-blend** | 72.4±2.3 | 71.5±1.8 | **74.2±1.2** | 59.8±1.7 | 59.2±1.2 | **62.1±0.6** |
| | **Adapt-patch** | 2.2±0.7 | 6.6±0.5 | **14.3±0.3** | 10.9±2.3 | 13.4±1.4 | **16.2±0.9** |

Table 5: Full Performance on Type I backdoor attacks (Cifar100).

| Model Defenses | Attacks | ResNet18 | | | ResNet18 → VGG16 | | |
|---|---|---|---|---|---|---|---|
| | | Radnom | FUS | Boundary | Radnom | FUS | Boundary |
| No defense | BadNet | 82.8±2.3 | 84.1±1.5 | 78.1±0.9 | 83.1±2.6 | 86.3±1.9 | 80.4±1.2 |
| | Blend | 82.7±2.6 | 83.9±1.7 | 77.9±1.1 | 79.6±2.8 | 82.9±2.1 | 75.2±1.3 |
| | Adapt-blend | 67.1±1.9 | 69.2±1.3 | 64.5±0.7 | 70.6±2.4 | 74.1±1.5 | 69.3±0.9 |
| | Adapt-patch | 78.2±1.2 | 81.4±1.4 | 75.1±0.8 | 82.4±2.7 | 86.7±1.8 | 83.1±1.1 |
| SS | BadNet | 0.6±0.2 | 3.7±1.3 | **6.5±0.8** | 0.7±0.2 | 4.5±1.8 | **6.9±0.9** |
| | Blend | 0.7±0.3 | 2.6±1.5 | **5.2±1.1** | 1.6±0.7 | 3.5±1.1 | **5.7±0.5** |
| | Adapt-blend | **7.3±1.7** | 4.8±1.3 | 5.7±0.7 | 12.8±1.9 | 11.7±1.3 | **15.6±0.7** |
| | Adapt-patch | 9.5±2.1 | 10.9±1.7 | **14.2±1.2** | 10.5±2.1 | 11.3±1.2 | **14.9±0.3** |
| AC | BadNet | 0.4±0.1 | 7.5±1.2 | **10.1±0.6** | 2.6±0.9 | 8.2±1.6 | **11.4±1.1** |
| | Blend | 0.2±0.1 | 9.3±2.3 | **11.9±1.7** | 3.4±1.5 | 7.6±1.2 | **9.7±0.8** |
| | Adapt-blend | 10.2±2.5 | 18.7±2.1 | **23.5±1.6** | 3.4±2.3 | 4.2±1.8 | **6.7±0.7** |
| | Adapt-patch | 13.5±2.1 | 21.7±1.3 | **26.8±1.0** | 5.2±1.6 | 5.7±1.2 | **7.4±0.9** |
| SCAn | BadNet | **85.5±3.8** | 84.9±3.2 | 83.2±2.1 | 78.3±2.9 | 77.6±2.1 | **81.9±1.4** |
| | Blend | **84.1±1.6** | 83.5±1.2 | 82.9±0.8 | 80.2±2.1 | **81.4±1.3** | 80.9±0.9 |
| | Adapt-blend | 69.7±2.7 | 68.7±1.8 | **72.6±1.1** | 68.8±3.4 | **69.4±1.6** | 67.9±1.5 |
| | Adapt-patch | 71.7±1.5 | 71.3±0.9 | **73.9±0.7** | 81.9±2.7 | 81.2±1.6 | **82.1±1.1** |
| STRIP | BadNet | 72.3±2.7 | 71.8±1.8 | **77.1±1.2** | 67.6±3.2 | 68.1±2.4 | **73.7±1.3** |
| | Blend | **83.2±3.2** | 82.9±2.5 | 82.8±1.6 | 71.9±2.7 | 71.2±1.6 | **75.1±0.9** |
| | Adapt-blend | 64.4±3.7 | 67.9±2.3 | **70.6±1.6** | 69.2±2.8 | **70.8±1.5** | 68.5±0.7 |
| | Adapt-patch | 67.8±2.5 | 67.5±1.7 | **72.7±1.3** | 74.7±1.9 | **75.4±1.3** | 73.5±0.8 |
| SPECTRE | BadNet | 0.2±0.1 | 3.9±1.4 | **7.3±0.6** | 0.6±0.2 | 2.5±0.7 | **4.1±0.5** |
| | Blend | 0.6±0.2 | 12.4±1.5 | **14.7±0.5** | 9.5±1.4 | 12.5±1.3 | **14.7±0.9** |
| | Adapt-blend | 14.8±1.5 | 19.6±1.3 | **20.3±0.9** | 15.7±2.3 | 16.9±1.7 | **20.1±1.2** |
| | Adapt-patch | 17.9±2.1 | 25.8±1.4 | **27.3±0.8** | 19.3±1.9 | 20.5±1.3 | **21.6±0.7** |
| ABL | BadNet | 9.3±2.4 | 13.9±1.7 | **17.4±0.7** | 5.7±1.3 | 9.6±1.5 | **10.2±1.1** |
| | Blend | 20.8±2.7 | 22.7±1.3 | **25.7±1.1** | 59.1±2.7 | 58.3±2.1 | **62.6±1.4** |
| | Adapt-blend | 23.7±2.5 | 23.2±1.5 | **25.8±0.8** | 43.3±3.2 | 44.8±2.7 | **46.4±1.6** |
| | Adapt-patch | 19.8±1.8 | 20.4±1.2 | **21.9±1.0** | 45.8±2.8 | 45.2±1.7 | **47.9±1.3** |
| FP | BadNet | 29.4±2.7 | 30.1±1.4 | **35.3±0.9** | 61.8±3.5 | 63.7±2.1 | **64.1±1.6** |
| | Blend | 67.2±2.8 | 68.1±2.3 | **71.1±1.1** | 73.1±2.9 | 72.7±1.8 | **74.2±1.3** |
| | Adapt-blend | 60.7±1.5 | 57.3±1.1 | **62.6±0.8** | 69.7±3.1 | 70.3±2.5 | **73.4±1.4** |
| | Adapt-patch | 66.3±2.4 | 64.1±1.9 | **69.7±1.2** | 70.1±2.5 | **69.7±1.8** | 69.5±1.5 |
| NC | BadNet | 35.6±3.4 | 42.1±2.9 | **52.4±1.4** | 43.7±3.2 | 44.8±2.5 | **49.5±0.8** |
| | Blend | 78.1±2.5 | **79.4±1.8** | 77.2±1.3 | 68.4±2.4 | 69.2±1.6 | **72.3±1.1** |
| | Adapt-blend | 66.9±1.7 | 64.2±1.3 | **70.3±0.9** | 66.2±2.7 | 65.4±1.4 | **67.8±0.6** |
| | Adapt-patch | 18.3±1.3 | 19.5±0.9 | **23.6±0.4** | 2.7±0.7 | 4.1±1.2 | **4.6±0.8** |

Table 6: Full Performance on Type II backdoor attacks.

| | Model Defense | Attacks | ResNet18 | | | ResNet18 → VGG16 | | |
|---|---|---|---|---|---|---|---|---|
| | | | Random | FUS | CBS | Random | FUS | CBS |
| CIFAR10 | No Defenses | Hidden-trigger | 81.9±1.5 | 84.2±1.2 | 76.3±0.8 | 83.4±2.1 | 86.2±1.3 | 79.6±0.7 |
| | | LC | 90.3±1.2 | 92.1±0.8 | 87.2±0.5 | 91.7±1.4 | 93.7±0.9 | 87.1±0.8 |
| | NC | Hidden-trigger | 6.3±1.4 | 5.9±1.1 | **8.7±0.9** | 10.7±2.4 | 11.2±1.5 | **14.7±0.6** |
| | | LC | 8.9±2.1 | 8.1±1.6 | 12.6±1.1 | 11.3±2.6 | 9.8±1.1 | 12.9±0.9 |
| | SS | Hidden-trigger | 68.5±3.2 | 69.3±2.4 | **74.1±1.3** | 75.7±3.1 | 74.8±2.3 | **76.2±1.1** |
| | | LC | **87.2±1.3** | 86.6±0.8 | 86.9±0.5 | 85.4±2.7 | **85.5±1.8** | 84.2±1.2 |
| | FP | Hidden-trigger | 11.7±2.6 | 9.9±1.3 | **14.3±0.9** | 8.6±2.4 | 8.1±1.4 | **11.8±0.8** |
| | | LC | 10.3±2.1 | 13.5±1.2 | **20.4±0.7** | 7.9±1.7 | 8.2±1.1 | **10.6±0.7** |
| | ABL | Hidden-trigger | 1.7±0.8 | 5.6±1.6 | **10.5±1.1** | 3.6±1.1 | 8.8±0.8 | **10.4±0.6** |
| | | LC | 0.8±0.3 | 8.9±1.5 | **12.1±0.8** | 1.5±0.7 | 9.3±1.2 | **12.6±0.8** |
| CIFAR100 | No Defenses | Hidden-trigger | 80.6±2.1 | 84.1±1.8 | 78.9±1.3 | 78.2±2.3 | 81.4±1.6 | 75.8±1.2 |
| | | LC | 86.3±2.3 | 87.2±1.4 | 84.7±0.9 | 84.7±2.8 | 85.2±1.4 | 81.5±1.1 |
| | NC | Hidden-trigger | 3.8±1.4 | 4.2±0.9 | **7.6±0.7** | 4.4±1.1 | 5.1±1.2 | **6.8±0.9** |
| | | LC | 6.1±1.8 | 5.4±1.1 | **8.3±0.5** | 3.9±1.2 | 3.8±0.9 | **8.3±0.7** |
| | SS | Hidden-trigger | 72.5±2.6 | 71.9±1.7 | **74.7±1.2** | 75.3±3.1 | 74.8±2.1 | 73.1±1.3 |
| | | LC | **80.4±2.4** | 80.1±1.4 | 79.6±1.3 | 82.9±2.7 | **83.5±1.8** | 81.4±1.0 |
| | FP | Hidden-trigger | 15.3±3.1 | 16.7±0.9 | **18.2±0.7** | 8.9±1.3 | 9.3±1.1 | **10.3±0.7** |
| | | LC | 13.8±2.7 | 12.7±1.5 | **14.9±0.6** | 10.3±1.4 | 9.9±0.8 | **12.2±0.5** |
| | ABL | Hidden-trigger | 2.3±0.9 | 3.9±1.3 | **6.5±1.1** | 3.7±0.9 | 3.5±0.7 | **6.4±0.4** |
| | | LC | 0.9±0.2 | 2.7±0.8 | **6.2±1.2** | 2.5±0.8 | 2.1±0.7 | **6.7±0.5** |

Table 7: Performance on Type III backdoor attacks.

| | Model Defense | Attacks | ResNet18 Random | FUS | CBS | VGG16 Random | FUS | CBS |
|---|---|---|---|---|---|---|---|---|
| **CIFAR10** | **No Defenses** | Lira | 91.5±1.4 | 92.9±0.7 | 88.2±0.8 | 98.3±0.8 | 99.2±0.5 | 93.6±0.4 |
| | | WaNet | 90.3±1.6 | 91.4±1.3 | 87.9±0.7 | 96.7±1.4 | 97.3±0.9 | 94.5±0.5 |
| | | WB | 88.5±2.1 | 90.9±1.9 | 86.3±1.2 | 94.1±1.1 | 95.7±0.8 | 92.8±0.7 |
| | **AC** | Lira | 90.7±2.1 | 90.8±1.4 | **91.1±0.9** | 90.5±3.1 | 89.8±2.3 | **91.2±1.2** |
| | | WaNet | **90.5±1.3** | 89.6±0.9 | 89.9±0.6 | 90.8±3.5 | **91.5±2.1** | 90.4±1.4 |
| | | WB | 87.1±2.3 | 87.7±1.5 | **88.2±1.3** | 90.4±2.8 | 89.5±1.7 | **91.1±0.9** |
| | **SS** | Lira | 86.5±2.7 | 89.6±1.6 | **90.1±1.3** | 90.5±2.5 | **91.3±1.6** | 90.1±1.1 |
| | | WaNet | 87.4±3.1 | **89.4±1.5** | 88.2±1.4 | 90.6±2.6 | 90.8±1.1 | **91.2±0.7** |
| | | WB | 86.4±2.8 | 86.1±2.3 | **88.1±1.7** | 87.6±3.2 | 88.2±2.5 | **89.9±1.3** |
| | **NC** | Lira | 10.3±1.6 | 12.5±1.1 | **16.1±0.7** | 14.9±1.5 | 18.3±1.1 | **19.6±0.8** |
| | | WaNet | 8.9±1.5 | 10.1±1.3 | **13.4±0.9** | 10.5±1.1 | 12.2±0.7 | **13.7±0.9** |
| | | WB | 20.7±2.1 | 19.6±1.2 | **27.2±0.6** | 23.1±1.3 | 24.9±0.8 | **28.7±0.5** |
| | **STRIP** | Lira | 81.5±3.2 | 82.3±2.3 | **87.7±1.1** | 82.8±2.4 | 81.5±1.7 | **84.6±1.3** |
| | | WaNet | 80.2±3.4 | 79.7±2.5 | **86.5±1.4** | 77.6±3.1 | **79.3±2.2** | 78.2±1.5 |
| | | WB | 80.1±2.9 | 81.7±1.8 | **86.6±1.2** | 83.4±2.7 | 82.6±1.8 | **87.3±1.1** |
| | **FP** | Lira | 6.7±1.7 | 6.2±1.2 | **12.5±0.7** | 10.4±1.1 | 9.8±0.8 | **13.3±0.6** |
| | | WaNet | 4.8±1.3 | 6.1±0.9 | **8.2±0.8** | 6.8±0.9 | 6.4±0.6 | **8.3±0.4** |
| | | WB | 20.8±2.3 | 21.9±1.7 | **28.3±1.1** | 25.7±1.3 | 26.2±1.2 | **29.1±0.7** |
| **CIFAR100** | **No Defenses** | Lira | 98.2±0.7 | 99.3±0.2 | 96.1±1.3 | 97.1±0.8 | 99.3±0.4 | 94.5±0.5 |
| | | WaNet | 97.7±0.9 | 99.1±0.4 | 94.3±1.2 | 96.3±1.2 | 98.7±0.9 | 94.1±0.7 |
| | | WB | 95.1±0.6 | 96.4±1.1 | 94.7±0.9 | 93.2±0.9 | 96.7±0.4 | 91.9±0.8 |
| | **AC** | Lira | 83.5±2.6 | 82.4±1.9 | **87.1±1.3** | 85.2±2.8 | **85.7±2.1** | 84.2±1.2 |
| | | WaNet | 82.7±2.8 | 82.1±2.1 | **86.3±0.9** | 83.8±3.1 | 84.2±1.8 | **85.1±0.9** |
| | | WB | 83.2±2.4 | 84.9±1.6 | **90.2±1.2** | 90.5±2.4 | 89.3±1.5 | **91.8±0.9** |
| | **SS** | Lira | 93.2±1.7 | **94.6±1.3** | 92.8±0.8 | 91.8±1.9 | 90.7±1.3 | **92.1±0.7** |
| | | WaNet | 92.4±1.9 | **93.3±1.0** | 92.7±0.6 | **90.5±2.3** | 90.1±1.4 | 90.3±1.1 |
| | | WB | 92.9±1.3 | 92.7±0.8 | **94.1±0.9** | 90.1±2.1 | 90.4±1.6 | **92.5±0.8** |
| | **NC** | Lira | 0.2±0.1 | 1.7±1.2 | **5.8±0.9** | 3.4±0.7 | 3.9±1.0 | **5.2±0.9** |
| | | WaNet | 1.6±0.8 | 3.4±1.3 | **5.2±0.8** | 2.9±0.6 | 2.5±0.8 | **4.1±1.2** |
| | | WB | 7.7±1.5 | 7.5±0.9 | **13.7±0.7** | 8.5±1.3 | 7.6±0.9 | **11.9±0.7** |
| | **STRIP** | Lira | 84.3±2.7 | 83.7±1.5 | **87.2±1.1** | 82.7±2.5 | 83.4±1.8 | **83.8±1.4** |
| | | WaNet | 82.5±2.4 | 82±1.6 | **83.9±0.9** | 81.4±2.7 | **82.5±1.7** | 82.0±0.8 |
| | | WB | 85.8±1.9 | 86.4±1.2 | **88.1±0.8** | 82.9±2.4 | 82.3±1.5 | **84.5±1.4** |
| | **FP** | Lira | 87.4±1.9 | 88.2±1.1 | **89.9±0.9** | 82.5±3.2 | 81.8±2.4 | **86.7±1.1** |
| | | WaNet | 86.7±1.7 | 86.3±0.9 | **89.3±0.7** | 81.7±2.9 | 82.1±1.8 | **85.6±1.3** |
| | | WB | 89.2±1.5 | 89.7±0.7 | **92.1±0.5** | 83.6±2.4 | 83.3±1.7 | **87.9±0.8** |

