# OpenReview forum: "Confidence-driven Sampling for Backdoor Attacks"
_ICLR.cc/2024/Conference — Submitted to ICLR 2024_

### Official Review · Reviewer_ycqn · 2023-10-16

**Soundness:** 3 good
**Presentation:** 3 good
**Contribution:** 3 good
**Rating:** 6
**Confidence:** 4

**Summary:**

Traditional approaches proposed to mount backdoor attacks inject triggers into training samples selected at random. This paper presents a new approach for selecting the training samples to poison in order to launch a more effective backdoor attack. To that end, the authors select samples that are close to the decision boundary between different classes. These samples are the ones that have a low confidence (measured as the probability of a particular class).

The authors test their approach against 2 baselines: a random baseline, and a baseline where poisoned samples are chosen based on how early they are forgotten during training (FUS). For each of these choices, the authors implement several backdoor attacks, including BadNets, Blend and Adaptive Attacks. The attacks are evaluated when no defenses are employed, and when several defenses are used, such as SS, STRIP, ABL and NC.

The results on CIFAR10 and CIFAR100 show that selecting the samples to poison is more effective than poisoning random samples. Furthermore, several attacks that are not effective (when defenses are employed) with random sampling become effective when the samples to-poison are chosen.

Finally, the authors investigate the choice of close-to-boundary and far-from-boundary samples to poison to validate that poisoning samples close to the decision boundary is more effective.

**Strengths:**

- the paper presents a new approach for selecting the samples to be poisoned during a backdoor attack, as opposed to random sampling. the results of the paper show that a good selection is more effective than random selection
- the paper evaluates several backdoor attacks and defenses when random and targeted sampling are used

**Weaknesses:**

- the paper considers 2 small datasets: cifar10 and cifar100. it would be great to evaluate on larger datasets, such as imagenet
- the paper evaluates defenses that rely on some notion of outlier detection, and as such, the selection mechanism is effective. the authors however do not evaluate defenses that are not based on outlier detection, such as [1]


[1] Provable Guarantees against Data Poisoning Using Self-Expansion and Compatibility, Jin et al., 2021

**Questions:**

- FUS is not a method proposed for selecting poisoned samples. how do you do the selection for this technique?
- can you evaluate your method on larger datasets such as imagenet?

---

> ### Author Response · Authors · 2023-11-15
> **Responses to Reviewer ycqn**
>
> We thank the reviewer for raising these concerns. We will address these concerns by discussing the following questions: (1) evaluation on larger datasets such as ImageNet; (2) evaluation on defenses that do not depend on outlier detection; (3) clarification for FUS.
>
> **Q1: Evaluation on larger datasets such as ImageNet.**
>
> We thank the reviewer for this suggestion. We conduct experiments on Tiny-ImageNet. Success rates of random sampling and CBS combined with 3 types of attacks and against various defenses are shown in the following tables. These results demonstrate that CBS is applicable to larger datasets and consistently improves the stealthiness of different attacks.
>
> Performance on Type I backdoor attacks:
> |               | Attacks        | random | CBS  |
> |---------------|----------------|--------|------|
> | No defenses   | BadNet         | 89.5   | 83.1 |
> |               | Blended        | 83.4   | 81.6 |
> |               | Adaptive-Blend | 67.2   | 66.2 |
> |               | Adaptive-Patch | 84.5   | 81.7 |
> | SS            | BadNet         | 0.4    | 18.5 |
> |               | Blended        | 37.2   | 46.3 |
> |               | Adaptive-Blend | 59.4   | 65.1 |
> |               | Adaptive-Patch | 75.3   | 78.5 |
> | Strip         | BadNet         | 0.6    | 12.2 |
> |               | Blended        | 46.2   | 54.6 |
> |               | Adaptive-Blend | 58.4   | 63.3 |
> |               | Adaptive-Patch | 69.5   | 72.8 |
>
> Performance on Type II backdoor attacks:
> |               | Attacks        | random | CBS  |
> |---------------|----------------|--------|------|
> | No defenses   | Hidden-trigger | 59.7   | 54.3 |
> | NC            | Hidden-trigger | 5.3    | 11.5 |
> | FP            | Hidden-trigger | 8.4    | 12.1 |
> | ABL           | Hidden-trigger | 1.8    | 4.2  |
>
> Performance on Type III backdoor attacks:
> |               | Attacks        | random | CBS  |
> |----------------|---------|-------|------|
> | No defenses | WaNet | 98.5 | 96.1 |
> |             | LiRA  | 99.3 | 96.4 |
> |             | WB    | 98.2 | 92.7 |
> | NC          | WaNet | 5.4  | 10.7 |
> |             | LiRA  | 6.3  | 10.2 |
> |             | WB    | 9.6  | 15.1 |
> | FP          | WaNet | 9.5  | 13.6 |
> |             | LiRA  | 8.7  | 13.8 |
> |             | WB    | 10.2 | 16.5 |
>
>
> **Q2: Evaluatuion on defenses that do not depend on outlier detection.**
>
> We would like to politely point out that we evaluate defenses that do not depend on outlier detections such as Neural Cleanse[3], Fine Pruning[4], Anti-Backdoor Learning[5], in this work. We present partial results in the main text Table 1-3 and full results in the Appendix Table 4-7. According to these results, our method can also effectively improve the attacks' resistance against non-outlier-detection defenses. We are grateful for the defense in [1] pointed out by the reviewer. We notice that the authors published an updated version in [2] so we follow the latest paper. In detail, we test on CIFAR-10 with model ResNet18, and evaluate the performance of attacks BadNet and Clean-label attack. We compare our method with random sampling. We present both success rate (ASR) and clean accuracy (Clean Acc) in the following Table. We include another defense Neural Cleanse for a convenient comparison. The results show that under this powerful defense, our CBS can still significantly improve the stealthiness of backbone methods.
>
> | Attacks |      | Random |       | CBS   |       |
> |---------|------|--------|-------|-------|-------|
> |         |      | ASR    | Clean Acc | ASR  | Clean Acc |
> | ISPL+B  | BadNet | 0.3    | 93.1    | 13.1 | 93.3      |
> |         | LC     | 0.9    | 92.2    | 10.3 | 92.4      |
> | NC      | BadNet | 1.1    | 93.5    | 24.6 | 93.1      |
> |         | LC     | 8.9    | 92.7    | 12.6 | 92.6      |
>
> **Q3: Clarification for FUS**
>
> As mentioned in the Introduction of the original paper, FUS [2] is a method to improve the efficiency of poisoning by a rational selection of poisoned samples. We follow the strategy in Sections 3.3, 4.1 and 4.2 in the original paper where they compare FUS with random sampling for fixed poison rates (which is referred to as mixing ratio in the original paper). In this case, we can compare it with random sampling and CBS for fixed poisoned rates.
>
> **Additional comments:**
> We follow the reviewer's suggestion and conduct experiments on Tiny-ImageNet.
>
> We hope our response can address your concerns. We are grateful for your valuable reviews and look forward to further feedback.
>
> **References**
>
> [1] Provable Guarantees against Data Poisoning Using Self-Expansion and Compatibility
>
> [2] Data-Efﬁcient Backdoor Attacks
>
> [3] Neural Cleanse: Identifying and Mitigating Backdoor Attacks in Neural Networks
>
> [4] Fine-Pruning: Defending Against Backdooring Attacks on Deep Neural Networks
>
> [5] Anti-Backdoor Learning: Training Clean Models on Poisoned Data

---

> ### Author Response · Authors · 2023-11-19
> **A friendly reminder**
>
> We are grateful for your valuable comments. We hope that our responses have addressed your concerns. If you have any further concerns, please let us know. We are looking forward to hearing from you.

---

> > ### Comment · Reviewer_ycqn · 2023-11-19
> >
> > Yes, thank you for running the additional experiments and for clarifying the use of FUS!

---

> > > ### Author Response · Authors · 2023-11-20
> > >
> > > We are glad to hear that! Thank you for your feedback.

---

### Official Review · Reviewer_gKjE · 2023-10-27

**Soundness:** 3 good
**Presentation:** 4 excellent
**Contribution:** 2 fair
**Rating:** 5
**Confidence:** 4

**Summary:**

The authors propose a new sampling procedure for backdoor attacks. Standard backdoor attacks poison (i.e., insert a backdoor trigger on) *random* examples from the training set. The authors claim that strategically choosing which training examples to poison can improve on the efficacy of a variety of attacks.  In particular, the authors choose to poison training examples on which a surrogate model exhibits low confidence. Then, the authors provide strong experimental evidence that this sampling scheme improves existing attacks. Additionally, the authors provide intuition for why their method might work on neural networks by analyzing an SVM model on a toy dataset.

**Strengths:**

- the authors propose a simple method with few hyperparameters that reliably gives good experimental results
- the proposed method can be combined with existing attacks
- the choice of backdoor attacks in the evaluation is thorough
- the writing is clear and concise

**Weaknesses:**

My main concern is about how practical the idea is. In particular, the adversary is assumed to have:
- *edit* access to the entire train set
- access to a surrogate model that is similar to the model that is being backdoored.

For example, suppose I'm an (adversarial) user of some social media platform P. I know P will train *some* model on the data of its users (including my own data). I could execute a backdoor attack by simply inserting a backdoor trigger in each of my datapoints (e.g., images). However, to execute the proposed attack, I would need the ability to insert data into arbitrary users' profiles---this already makes the attack a lot harder to execute. Additionally, I would need access to a model that acts similarly to the model P will train. Because of this, now I need knowledge of the model P aims to train---in my view, another significant challenge in practice.

As another example in the same vein, consider the setup in [1]. There, the authors poison *expired* web-pages. The above two challenge persists in this setup.

Additional weakness:
- Evaluated defenses: it makes (intuitive) sense to me that the proposed attack works well against outlier-based defenses. However, it is unclear to me whether this will translate to defenses based on model behavior like [2] and [3].


[1] Carlini, Nicholas, Matthew Jagielski, Christopher A. Choquette-Choo, Daniel Paleka, Will Pearce, Hyrum Anderson, Andreas Terzis, Kurt Thomas, and Florian Tramèr. "Poisoning web-scale training datasets is practical." arXiv preprint arXiv:2302.10149 (2023).

[2] Jin, Charles, Melinda Sun, and Martin Rinard. "Incompatibility Clustering as a Defense Against Backdoor Poisoning Attacks." In The Eleventh International Conference on Learning Representations. 2022.

[3] Khaddaj, Alaa, Guillaume Leclerc, Aleksandar Makelov, Kristian Georgiev, Hadi Salman, Andrew Ilyas, and Aleksander Madry. "Rethinking Backdoor Attacks." (2023). https://arxiv.org/abs/2307.10163

**Questions:**

- in Table 1, why does your method reduce the efficacy of the attack in the absence of a defense?

---

> ### Author Response · Authors · 2023-11-15
> **Responses to Reviewer gKjE (1/2)**
>
> We thank the reviewer for raising these concerns. We address these concerns by discussing the following questions: (1) Does the attacker have "edit access" to the entire training set? Is this assumption practical? (2) Does the attacker have access to the surrogate model? Is this scenario practical? (3) discuss the defenses mentioned by the reviewer; (4) Why does the performance drop in the absence of defenses in Table 1?
>
> **Q1: Does the attacker have "edit access" to the entire training set? Is this assumption practical?**
>
> We acknowledge that there exist some practical scenarios when the attacker has edit access to the entire training set. For example, the attacker can be a model provider or a training platform provider as described in scenarios 1 and 2 in [1]. Under these scenarios, the attacker have the capacity to modify any data in the training set as he wishes. Therefore, our method can be easily adapted for selection to improve the stealthiness of the backdoor.
>
> Moreover, our work does not necessarily assume the attacker has full edit access to the training set.
> When only allowed to insert triggers to part of the training data, the attacker can still select samples from his own data via our proposed method while keeping other users' data clean if the attacker is concerned about the stealthiness of the backdoor. To validate the feasibility of our proposed method under this scenario, we conducted an additional experiment on CIFAR-10. We allocated 10% of the training set as the attacker's private data, and the attacker is not allowed to perturb the rest 90%. We assume that the attacker selects 10% from his own data to insert triggers. To apply our method, we train a ResNet18 as the surrogate model and select samples from the private data. We evaluated the poisoning effect on both ResNet18 and VGG16. We considered the BadNet attack and defenses such as Spectral Signature(SS), Strip, and Neural Cleanse(NC). Our results (success rates), detailed in the following Table, suggest that our approach can indeed enhance the stealth of backbone attacks, even when the full training set is not accessible.
>
> | Defenses    | Model   | Random | CBS  |
> |-------------|---------|--------|------|
> | No defenses | ResNet18| 99.9   | 93.7 |
> |             | VGG16   | 99.8   | 95.2 |
> | SS          | ResNet18| 1.2    | 15.7 |
> |             | VGG16   | 3.5    | 10.4 |
> | NC          | ResNet18| 2.7    | 14.4 |
> |             | VGG16   | 3.6    | 12.8 |
> | Strip       | ResNet18| 1.5    | 13.2 |
> |             | VGG16   | 1.3    | 10.9 |
>
> **Q2: Does the attacker have access to the surrogate model? Is this scenario practical?**
>
> **First**, we would like to clarify that there exist some practical scenarios where surrogate models are available. When the attacker is a model provider or a training platform provider, he has access to the model information and thus can easily utilize a surrogate model for selection via our proposed method.
>
> **Second**, we would like to highlight that it is a common assumption to have a surrogate model. The previous works in [2][3][4] also consider poisoning attacks in the same setting to leverage a surrogate model for generating poisoning samples. Notably, the architecture or parameters of the surrogate model do not need to be the same as the victim model. In our work, in Table 1 and 2, we present experiments where poisons are generated from a ResNet18 model, and we test the poisoning performance on the victim model VGG16. These results show that poisons can be transferred to different models. Existing work such as [7] also provides empirical evidence for the effectiveness of surrogate models under the black-box setting. In Tabel 4 in [7], poisons are generated from the surrogate model ResNet18 and transferred to MonileNet-V2, VGG11. The poisons are still effective. Therefore, we think the usage of surrogate models is practical.

---

> ### Author Response · Authors · 2023-11-15
> **Responses to Reviewer gKjE (2/2)**
>
> **Q3: Discuss the defenses mentioned by the reviewer.**
>
> We are grateful for the defenses highlighted by the reviewer. To begin, we would like to clarify that our evaluation includes not only outlier-based defenses but also non-outlier-based ones such as NC, ABL, and FP. While we present a portion of these results in the main paper (Tables 1-3), comprehensive details can be found in the Appendix (Tables 4-7). According to these results, our method can improve the attacks’ resistance against both types of defenses, and we do not observe a drop in performance against non-outlier-detection defenses.
>
> Regarding the defense method mentioned in [5], we encountered significant challenges in implementing it due to its complexity. The method requires training an extensive number of models (approximately 100,000) on randomly selected 50% subsets of the poisoned dataset. This process is exceedingly time-intensive. Thus, we focus our experimental efforts on the defense proposed in [6]. In detail, we test on CIFAR-10 with model ResNet18, and evaluate the performance of attacks BadNet and Clean-label attack. We compare our method with random sampling. We present both success rate (ASR) and clean accuracy (Clean Acc) in the following Table. We include another defense Neural Cleanse for a convenient comparison. The results show that under this powerful defense, our CBS can still significantly improve the stealthiness of backbone methods.
>
> | Attacks |      | Random |       | CBS   |       |
> |---------|------|--------|-------|-------|-------|
> |         |      | ASR    | Clean Acc | ASR  | Clean Acc |
> | ISPL+B  | BadNet | 0.3    | 93.1    | 13.1 | 93.3      |
> |         | LC     | 0.9    | 92.2    | 10.3 | 92.4      |
> | NC      | BadNet | 1.1    | 93.5    | 24.6 | 93.1      |
> |         | LC     | 8.9    | 92.7    | 12.6 | 92.6      |
>
> **Q4: Why does the performance drop in the absence of defenses in Table 1?**
>
> Intuitively, our method selects samples close to the boundary, and this targeted selection results in a more constrained selection space compared to random sampling, which may lead to a slightly reduced performance in the absence of defenses. For instance, when high-confidence samples for their true classes are used as poisoning samples, there is a significant shift in the decision boundary, typically leading to a higher success rate. Moreover, in Section 4.3, we theoretically illustrate that boundary samples (with lower confidence) have a smaller success rate than samples that are further from the boundary (with higher confidence), which indicates that our methods slightly compromise the performance without defenses. Our analysis and experimental results show that there exists a trade-off between clean performance and stealthiness, and our method is stronger against defenses.
>
> We hope our responses can address the reviewer's concerns. We are grateful for the insightful reviews and look forward to further interactions.
>
> **References**
>
> [1] Backdoor Learning: A Survey
>
> [2] Hidden Trigger Backdoor Attacks
>
> [3] Label-Consistent Backdoor Attacks
>
> [4] Backdoor Attack with Imperceptible Input and Latent Modification
>
> [5] Rethinking Backdoor Attacks
>
> [6] Incompatibility Clustering as a Defense Against Backdoor Poisoning Attacks
>
> [7] Sleeper Agent: Scalable Hidden Trigger Backdoors for Neural Networks Trained from Scratch

---

> ### Author Response · Authors · 2023-11-19
> **A friendly reminder**
>
> We are grateful for your valuable comments. We hope that our responses have addressed your concerns. If you have any further concerns, please let us know. We are looking forward to hearing from you.

---

> ### Author Response · Authors · 2023-11-21
> **A kind reminder**
>
> We appreciate your reviews. We hope that our responses have adequately addressed your concerns. As the deadline for open discussion nears, we kindly remind you to share any additional feedback you may have.  We are keen to engage in further discussion.

---

> > ### Comment · Reviewer_gKjE · 2023-11-22
> > **Response**
> >
> > I appreciate the authors' rebuttal. I find the experiment targeting only 10% of the training data very promising. However, I still believe the proposed attack introduces significant challenges in practice and thus keep my score.

---

> > > ### Author Response · Authors · 2023-11-22
> > > **Response to Reviewer gKjE**
> > >
> > > We thank the reviewer's response and we respect the reviewer's opinion. For the reviewer's remaining concern, we would provide a further explanation of why we believe our method is practical for the attacker to select the samples to poison.
> > >
> > > **1.** We will discuss this issue based on the different roles of an attacker:
> > > * **A model provider**. Existing attacking methods such as clean-label [1], WaNet [2], LiRA [3], WB [4] all follow this scenario, and assume that adversary has full knowledge of the model architecture and training procedure, while also having access to the underlying data distribution (directly from Section 2.2 in [1]). Under this assumption, the attacker is usually considered as the model trainer or provider. For example, tech companies can provide facial recognition services for users; autonomous vehicle technology companies provide road sign recognition systems for third-party automobile manufacturers. However, there is a chance that these served models have backdoors inserted.
> > > * **A data uploader with limited permission to perturb samples.** Consider that a victim model collects data samples from various internet sources to train an ML model, and this dataset also includes samples uploaded by an attacker $A$. However, due to the platform limitations, the attacker $A$ can only upload 100 samples to the internet, although they have a much larger collection of samples. In this case, the attacker has the opportunity to carefully select these 100 samples from their personal collection to maximize the effectiveness of their poisoning strategy.
> > >
> > > **2.** We claim that most of the existing backdoor attack methods assume access to the training set, and our method follows the same setting. We conducted a brief survey to demonstrate this fact. As shown in the following table, the columns of "Training set", "Model" and "Training" indicate the access to training data, model architecture and training procedure, respectively.
> > >
> > > |     Method  |Training set |  Model | Training  |
> > > |-|-|-|-|
> > > |     BadNet[5] |     yes  |     yes      |     yes         |
> > > |     Blend[6]                     |     part            |     no       |     no          |
> > > |     Adaptive-blend[7]            |     yes             |     no       |     no          |
> > > |     Clean-label[1]         |     yes             |     yes      |     yes         |
> > > |     Backdoor Embedding[8]        |     yes             |     yes      |     yes         |
> > > |     LiRA[3]                      |     yes             |     yes      |     yes         |
> > > |     Hidden-backdoor[9]    |     yes             |     yes      |     no          |
> > > |     Sleeper agent[10]             |     yes             |     yes      |     no          |
> > > |     WaNet[2]                     |     yes             |     yes      |     yes         |
> > > |     Data efficient[11]            |     yes             |     yes      |     yes         |
> > > |     WB[4]      |     yes             |     yes      |     yes         |
> > > |     Input-aware[12]               |     yes             |     yes      |     yes         |
> > >
> > > **3.** We are grateful for the reviewer's positive feedback on our results for 10%. To further illustrate the practicality of our method, we conduct experiments for even smaller control rates as 5% and 1%. We compare our method with random sampling on the CIFAR-10 dataset and the ResNet18 model with backbone attack BadNet. The results demonstrate that our method consistently enhances the performance of the backbone attack against defenses when controlling smaller subsets. This further indicates the applicability of our approach in practical scenarios where the attacker has limited access to the training data, thus illustrating the practicality of our method.
> > >
> > >
> > > | Defenses    | Subset rate | Random | CBS  |
> > > |-------------|-------------|--------|------|
> > > | No defenses | 5%          | 98.4   | 92.9 |
> > > |             | 1%          | 97.2   | 92.3 |
> > > | SS          | 5%          | 0.9    | 12.4 |
> > > |             | 1%          | 0.6    | 8.5  |
> > > | NC          | 5%          | 1.4    | 10.7 |
> > > |             | 1%          | 2.2    | 8.4  |
> > > | Strip       | 5%          | 0.9    | 9.5  |
> > > |             | 1%          | 0.5    | 7.2  |
> > >
> > > **References**
> > >
> > > [1] Label-Consistent Backdoor Attacks
> > >
> > > [2] WaNet -- Imperceptible Warping-based Backdoor Attack
> > >
> > > [3] LIRA: Learnable, Imperceptible and Robust Backdoor Attacks
> > >
> > > [4] Backdoor Attack with Imperceptible Input and Latent Modification
> > >
> > > [5] BadNets: Identifying Vulnerabilities in the Machine Learning Model Supply Chain
> > >
> > > [6] Targeted Backdoor Attacks on Deep Learning Systems Using Data Poisoning
> > >
> > > [7] Revisiting the Assumption of Latent Separability for Backdoor Defenses
> > >
> > > [8] Backdoor Embedding in Convolutional Neural Network Models via Invisible Perturbation
> > >
> > > [9] Hidden Trigger Backdoor Attacks
> > >
> > > [10] Sleeper Agent: Scalable Hidden Trigger Backdoors for Neural Networks Trained from Scratch
> > >
> > > [11] Data-Efﬁcient Backdoor Attacks
> > >
> > > [12] Input-Aware Dynamic Backdoor Attack

---

### Official Review · Reviewer_okLZ · 2023-10-29

**Soundness:** 3 good
**Presentation:** 3 good
**Contribution:** 3 good
**Rating:** 6
**Confidence:** 4

**Summary:**

The paper explores the strategies of choosing samples to inject triggers for training backdoor models. To improve the robustness against backdoor defenses, the paper proposes a confidence-driven sampling strategy for backdoor attacks. Specifically, the proposed method chooses samples with lower confidence scores to inject triggers, making the backdoored models difficult to be defended. Extensive experiments show the effectiveness of proposed method.

**Strengths:**

To improve the robustness against backdoor defenses, the paper proposes a simple and effective sampling choosing strategy for backdoor attacks. Also, the proposed method can be integrated into various backdoor attacks to improve the robustness. The paper analyzes the proposed method theoretically, indicating the rationality of proposed method.

**Weaknesses:**

The method is not efficient because it needs to train a suggorate model (e.g. ResNet18) to check the confidence scores. In the paper, the used datasets are two small-scale datasets including CIFAR10 and CIFAR100. Is it time-consuming when traing a suggorate model on a large dataset e.g. image-net?

In the experiments (Table 1, 2 and 3), the used backdoor defenses are not up-to-date. It is better to compare with some recent backdoor attacks e.g. i-bau [1].

[1] Zeng, Yi, et al. "Adversarial Unlearning of Backdoors via Implicit Hypergradient." International Conference on Learning Representations. 2021.

**Questions:**

Does the proposed method perform better against defenses depending on outlier detection than other backdoor defense methods?

In Figure 1, the paper shows the visualizations of two dirty-label attacks. Is there same observations for clean-label attacks?

In Table 1, 2 and 3, the proposed method achieves lower ASRs compared to Random and FUS under the condition of "No Defenses". Could the authors provide more explanations about the results?

---

> ### Author Response · Authors · 2023-11-15
> **Responses to Reviewer okLZ (1/2)**
>
> We thank the reviewer for raising these concerns. To address these concerns, we will discuss the following questions: (1) Is the proposed method inefficient, especially for training surrogate models on large datasets; (2) Evaluations on some recent backdoor attacks e.g. i-bau; (3) Does the proposed method perform better against defenses depending on outlier detection than other backdoor defense methods; (4) Are there same observations for clean-label attacks as for dirty-label attacks in Figure 1; and (5) Why the performance drop when no defenses.
>
> **Q1: Is the proposed inefficient, especially for training surrogate models on large datasets?**
>
> We would like to clarify that the proposed method does not suffer much from the time efficiency problem. **First**, we do not need to train the surrogate model till the end, because shorter training epochs are sufficient for selection and have the added benefit of reducing the risk of the model overfitting to the training set which may reduce the poisoning effect. As mentioned in Section 5.1, we only train the surrogate model for 60 epochs. We provide an ablation study about training epochs for the surrogate model in the CIFAR-10 dataset and ResNet18 model in Table below.
> Epochs|20|60|80|100
> --------|----|--|---|-----
> No defense|94.4|93.6|93.2|92.1
> SS|14.6|23.2|22.9|23.6
> Strip|12.1|26.2|26.7|25.9
> NC|13.5|24.6|24.2|24.8
> ABL|10.8|31.3|31.5|30.7
>
> According to this table, when the surrogate model is trained for few epochs(20), we can not have a good estimation for boundary thus the performance is much decreased; when we train for more epochs(80), the performance is similar to 60 epochs.  **Second**, many existing attacking methods such as hidden-trigger[2] and clean-label[3], utilize a surrogate model. When applying the proposed method to these attacks, we can directly leverage surrogate models of these attacks for selection, thus avoiding inefficiency. Moreover, for some large datasets such as ImageNet, there exist pre-trained models that can be directly used by the proposed method. We conduct experiments on Tiny-ImageNet where the surrogate model is trained for 60 epochs within 2 hours. The results are shown in Q1 in the response for Reviewer ycqn, and our method can consistently improve the attack's resistance against various defenses.
>
> **Q2: Evaluations on some recent backdoor attacks e.g. i-bau.**
>
> We thank the reviewer for suggesting the defense, I-BAU[1]. We compare our method (CBS) and random sampling (Random) on the CIFAR-10 dataset and ResNet18 model. We test various attacks including BadNet, Blend, WaNet and Hidden-trigger. We present both success rate(ASR) and clean accuracy(Clean Acc, accuracy on samples without trigger) in the following Table.
>
> | Attacks        | Random |     | CBS  |     |
> |----------------|--------|-----|------|-----|
> |                | ASR    | Clean Acc | ASR | Clean Acc |
> | **I-BAU**      |        |     |      |     |
> | BadNet         | 2.3    | 91.3 | 9.5  | 91.4 |
> | Blend          | 3.2    | 90.7 | 8.6  | 90.5 |
> | WaNet          | 1.7    | 90.5 | 8.2  | 90.6 |
> | Hidden trigger | 0.8    | 90.6 | 8.1  | 90.4 |
> | **NC**         |        |     |      |     |
> | BadNet         | 1.1    | 93.5 | 24.6 | 93.1 |
> | Blend          | 82.5   | 93.7 | 81.7 | 94.0 |
> | WaNet          | 8.9    | 94.1 | 13.4 | 93.8 |
> | Hidden trigger | 6.3    | 92.7 | 8.7  | 93.5 |
>
> We include another defense Neural Cleanse[7] for convenient comparison. We notice that I-BAU is much more powerful than other defenses such as Neural Cleanse[6] but compromises the clean performance. However, according to the results, our method can still improve the stealthiness of the backbone methods under this powerful defense.

---

> ### Author Response · Authors · 2023-11-15
> **Responses to Reviewer okLZ (2/2)**
>
> **Q3. Does the proposed method perform better against defenses depending on outlier detection than other backdoor defense methods?**
>
> In this work, we consider various defenses including outlier-detection methods like Spectral Signature[4], Activation Clustering[5], SPECTRE[6], and other methods like Neural Cleanse[7], Fine Pruning[8], and Anti-Backdoor Learning[9]. We present part of the results in the main paper Table 1-3 and full results in Appendix Table 4-7. According to these results, our method can improve the attacks' resistance against both types of defenses, and we do not observe a drop in performance against non-outlier-detection defenses. For instance, in Table 1, BadNet+CBS can achieve a success rate of 24.6% against NC which is not an outlier-detection defense, while achieving 23.7% against STRIP which is an outlier-detection defense.
>
> **Q4: Are there same observations for clean-label attacks as for dirty-label attacks in Figure 1?**
>
> Yes, we have similar observations for clean-label attacks such as Clean-label attack in [3]. We will include visualizations for Clean-label attacks in the appendix. Existing work [9] provides additional evidence. As shown in the introduction and Figure 1 in [10], poison and clean samples consistently form two separate clusters in the latent space, even for the Clean-label attack[3].
>
> **Q5:Why the performance drop when no defenses?**
>
> Intuitively, our method selects samples close to the boundary, and this targeted selection results in a more constrained selection space compared to random sampling, which may lead to a slightly reduced performance in the absence of defenses. For instance, when high-confidence samples for their true classes are used as poisoning samples, there is a significant shift in the decision boundary, typically leading to a higher success rate. Moreover, in Section 4.3, we theoretically illustrate that boundary samples (with lower confidence) have a smaller success rate than samples that are further from the boundary (with higher confidence), which indicates that our methods slightly compromise the performance without defenses. Our analysis and experimental results show that there exists a trade-off between clean performance and stealthiness, and our method is stronger against defenses.
>
> We hope our responses can address the reviewer's concerns. We are grateful for the insightful reviews and look forward to further interactions.
>
> **References**
>
> [1] Adversarial Unlearning of Backdoors via Implicit Hypergradient
>
> [2] Hidden Trigger Backdoor Attacks
>
> [3] Label-consistent backdoor attacks.
>
> [4] Spectral Signatures in Backdoor Attacks
>
> [5] Detecting Backdoor Attacks on Deep Neural Networks by Activation Clustering
>
> [6] SPECTRE: Defending Against Backdoor Attacks Using Robust Statistics
>
> [7] Neural Cleanse: Identifying and Mitigating Backdoor Attacks in Neural Networks
>
> [8] Fine-Pruning: Defending Against Backdooring Attacks on Deep Neural Networks
>
> [9] Anti-Backdoor Learning: Training Clean Models on Poisoned Data
>
> [10] Revisiting the Assumption of Latent Separability for Backdoor Defenses

---

> ### Author Response · Authors · 2023-11-19
> **A friendly reminder**
>
> We are grateful for your valuable comments. We hope that our responses have addressed your concerns. If you have any further concerns, please let us know. We are looking forward to hearing from you.

---

> > ### Comment · Reviewer_okLZ · 2023-11-23
> >
> > Thank you for the rebuttal. Most of concerns have been addressed and I will keep the score.

---

### Meta-Review · Area_Chair_dQp3 · 2023-12-12

**Metareview:**

This submission received truly borderline reviews. The rebuttal addressed some of the concerns but there are still concerns by at least one of the reviewers. The main concern is in the practicality of the proposed setting, mainly the assumption of attacker having edit access to the entire train set, also assuming access to a surrogate model. The authors provided some details about some real scenarios that this may be practical but could not convince the reviewer. The AC looked at the paper in more detail and agree with the concerns raised by the reviewers. The original submission assumes the attacker has access to the whole training data and can choose which data to poison. This threat model is not very practical: the more practical setting that is usually used in SOTA works is that the attacker can sneak in some "additional" data points to the training pool by uploading data to the web or similar ways hoping for the victim to download and use them. Hence, the AC agrees that the assumption of attacker choosing some data from the original training data to poison is not practical. In the rebuttal period, the authors provided some additional experiments that use only 10% of the data, however, those experiments are on a limited setting, e.g., BadNet setting only. Elaborating more on these settings can improve this paper in a future submission. Moreover, the paper can benefit by running experiments on larger scale datasets like ImageNet instead of CIFAR in the original submission and Tiny-ImageNet in the rebuttal.

**Justification For Why Not Higher Score:**

The main concern is in the practicality of the proposed setting, mainly the assumption of attacker having edit access to the entire train set, also assuming access to a surrogate model. The authors provided some details about some real scenarios that this may be practical but could not convince the reviewer. The AC looked at the paper in more detail and agree with the concerns raised by the reviewers. The original submission assumes the attacker has access to the whole training data and can choose which data to poison. This threat model is not very practical: the more practical setting that is usually used in SOTA works is that the attacker can sneak in some "additional" data points to the training pool by uploading data to the web or similar ways hoping for the victim to download and use them. Hence, the AC agrees that the assumption of attacker choosing some data from the original training data to poison is not practical. In the rebuttal period, the authors provided some additional experiments that use only 10% of the data, however, those experiments are on a limited setting, e.g., BadNet setting only. Elaborating more on these settings can improve this paper in a future submission. Moreover, the paper can benefit by running experiments on larger scale datasets like ImageNet instead of CIFAR in the original submission and Tiny-ImageNet in the rebuttal.

**Justification For Why Not Lower Score:**

N/A

---

### Decision · Program_Chairs · 2024-01-16

Reject